# Origin and evolutionary landscape of *Nr2f* transcription factors across Metazoa

**Ugo Coppola[1], Joshua S. Waxman**[1,2]*

**1** Molecular Cardiovascular Biology Division and Heart Institute, Cincinnati Children's Research Foundation, Cincinnati, Ohio, United States of America, **2** Department of Pediatrics, University of Cincinnati College of Medicine, Cincinnati, Ohio, United States of America

\* joshua.waxman@cchmc.org

## Abstract

### Background

Nuclear Receptor Subfamily 2 Group F (Nr2f) orphan nuclear hormone transcription factors (TFs) are fundamental regulators of many developmental processes in invertebrates and vertebrates. Despite the importance of these TFs throughout metazoan development, previous work has not clearly outlined their evolutionary history.

### Results

We integrated molecular phylogeny with comparisons of intron/exon structure, domain architecture, and syntenic conservation to define critical evolutionary events that distinguish the *Nr2f* gene family in Metazoa. Our data indicate that a single ancestral eumetazoan *Nr2f* gene predated six main Bilateria subfamilies, which include single *Nr2f* homologs, here referred to as *Nr2f1/2/5/6*, that are present in invertebrate protostomes and deuterostomes, *Nr2f1/2* homologs in agnathans, and *Nr2f1*, *Nr2f2*, *Nr2f5*, and *Nr2f6* orthologs that are found in gnathostomes. Four cnidarian *Nr2f1/2/5/6* and three agnathan *Nr2f1/2* members are each due to independent expansions, while the vertebrate *Nr2f1/Nr2f2* and *Nr2f5/Nr2f6* members each form paralogous groups that arose from the established series of whole-genome duplications (WGDs). *Nr2f6* members are the most divergent *Nr2f* subfamily in gnathostomes. Interestingly, in contrast to the other gnathostome *Nr2f* subfamilies, *Nr2f5* has been independently lost in numerous vertebrate lineages. Furthermore, our analysis shows there are differential expansions and losses of *Nr2f* genes in teleosts following their additional rounds of WGDs.

### Conclusion

Overall, our analysis of *Nr2f* gene evolution helps to reveal the origins and previously unrecognized relationships of this ancient TF family, which may allow for greater insights into the conservation of Nr2f functions that shape Metazoan body plans.

**Data Availability Statement:** All relevant data are within the manuscript and its Supporting information files.

**Funding:** Work in the manuscript was supported by National Institutes of Health (nih.gov) grants

R01 HL141186 and R01 HL137766 to JSW and by American Heart Association (heart.org) postdoctoral fellowship 831018 to UC. The funders had no role in study design, data collection and analysis, decision to publish, or preparation of the manuscript.

## Introduction

Nuclear hormone receptors (NRs) form a large, ancient superfamily of transcription factors (TFs) found in all Metazoa [1]. While NR functions are often dictated by interactions with specific ligands, including steroids, thyroid hormones, and retinoids [2, 3], the ligands for many NRs, called orphan NRs, are still not known [4]. Nuclear Receptor Subfamily 2 Group F Members (Nr2fs), initially named Chicken ovalbumin upstream promoter-transcription factors (Coup-TFs) due to their ability to bind the COUP element of the ovalbumin gene [5–7], are some of the most highly studied orphan NRs. Despite an overall expansion of the NR superfamily [1, 2], invertebrate phyla appear to have predominantly retained a single *Nr2f* gene. Only one *Nr2f* member is present in the protostome *Drosophila melanogaster* (fly), early-branching deuterostome *Strongylocentrotus purpuratus* (sea urchin) [8, 9], and invertebrate chordates *Branchiostoma floridae* (amphioxus) and *Ciona robusta* (sea squirt) [10, 11]. However, the number of *Nr2f* genes in early-branching metazoans is presently less clear. In cnidarians, one *Nr2f* has been reported in *Hydractinia echinata* [12, 13], while multiple have been reported in *Nematostella* and *Hydra vulgaris* [14, 15]. In contrast to most invertebrates, vertebrates have exhibited a significant expansion of the *Nr2f* family, with gnathostomes having multiple *Nr2f* genes. Furthermore, teleosts possess additional *Nr2f* Ohnologs (duplicates originating from whole-genome duplication (WGD)) [16], most likely reflecting the additional WGDs that have occurred in the teleost lineage [17, 18].

Nr2f proteins are highly conserved at the sequence level throughout Metazoa [19]. From the N-terminus to the C-terminus, all Nr2f proteins have six domains (Fig 1): an A/B domain, which contains the activating function-1 (AF-1) domain; the C domain, which contains the DNA-binding domain (DBD); the D domain (a linker); the E domain, which is comprised of the ligand-binding domain (LBD) and an AF-2 domain; the F domain (C-terminal) [20]. While the A/B domains are the most divergent in sequence, strikingly, the DBDs and LBDs of Nr2f members even from distantly related species (e.g. fly, sea urchin, frog, zebrafish, mouse, and human) are ~94% identical [21]. The extremely high degree of conservation among several species implies the preservation of critical roles for Nr2f in development and differentiation [21, 22]. Moreover, requirements for *Nr2f* genes have been found in organs of all three germ layers during embryogenesis [14, 22]. For instance, the *Drosophila Nr2f* homolog, called *seven up* (*svp*), is required for retinal, dorsal vessel, and liver development [23, 24]. Furthermore, Nr2f TFs in vertebrates appear to both have acquired diverse and retained redundant functions. For instance, in mice, *Nr2f1* is predominantly required for neural development with a role in regulation of premigratory and migratory neural crest cells in the developing hindbrain [25, 26]. However, the mouse *Nr2f2* gene is required for differentiation of mesodermal derivatives, including atrial cardiomyocytes of the heart and venous endothelial cells [22, 27, 28]. An example of redundancy is zebrafish *nr2f1a* and *nr2f2*, which are both required for proper ventricular cardiomyocyte and cranial muscle specification [29].

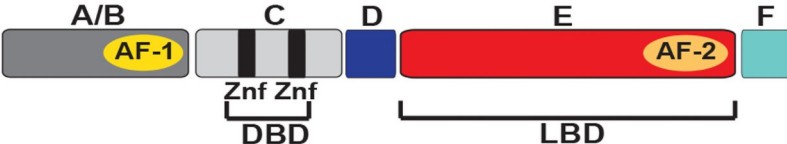

**Fig 1. Schematic of conserved domain architecture of Nr2f TFs.** A/B (N-terminal variable domain with transactivating AF-1 domain), C (DBD, which contains two Zinc finger (Znf) motifs), D (a linker domain), E (LBD plus transactivating AF-2 domain), and F (C-terminal).

While Nr2f proteins were initially identified as transcriptional activators of chicken ovalbumin gene [5], they have since been shown to function directly as both transcriptional activators and repressors in several developmental contexts [22, 30, 31]. Nr2fs can bind a range of different response elements [32–34] and in signaling reporter assays can compete with and inhibit retinoic acid receptors [35]. *In vivo* they bind numerous targets that reflect their various requirements in the specific tissues. For instance, *Nr2f1* KO mice also have inner ear defects [36]. In the mouse inner ear, direct targets of Nr2f1 include fatty acid binding protein 7 (FABP7), cellular retinoic acid binding protein 1 (CRABP1) [37], microRNA-140 (mi-R140), and Krüppel-like 9 (Klf9) [37, 38]. In adipogenesis, Nr2f2 directly represses peroxisome proliferator-activated receptor γ (PPARγ) downstream of canonical Wnt/β-catenin signaling [39]. In the mammalian heart, Nr2f2 is thought to directly orchestrate a regulatory network that facilitates atrial cardiomyocyte identity through concurrently promoting *Tbx5* and repressing *Irx4* and *Hey2*, the latter of which promote ventricular cardiomyocyte identity [40]. Thus, Nr2fs can activate and repress a range of direct targets related to their functions in specific tissues.

Despite the conservation and clear importance of this gene family to numerous developmental processes in Metazoa, we still do not completely understand the evolution of Nr2f TFs. Here, we investigated *Nr2f* family evolution through a combination of phylogenetic, domain architecture, intron/exon structure, and genomic synteny analyses. Our data show that the single *Nr2f* gene found in placozoans, represents the ancestral *Nr2f* to those found in cnidarians, protostomes, and deuterostomes. Importantly, a single *Nr2f* homolog, which we have named *Nr2f1/2/5/6*, is present in the majority of invertebrates, while most vertebrate genomes contain *Nr2f1*, *Nr2f2*, *Nr2f5*, and *Nr2f6* orthologs, which are derived from established rounds of WGDs [41, 42]. Interestingly, the invertebrate *Nr2f1/2/5/6* and agnathan *Nr2f1/2* homologs have retained the greatest similarity with vertebrate *Nr2f1* and *Nr2f2* paralogs. With respect to the vertebrate *Nr2f5* and *Nr2f6* paralogs, *Nr2f5* genes have been independently lost in some cartilaginous fish and amniote lineages, while the *Nr2f6* subfamily is the most divergent with respect to sequence and genomic structure. Overall, our data clarify the relationships among *Nr2f* genes within Metazoa and define the expansion, divergence, and independent loss of extant *Nr2f* genes in vertebrates, which will allow us to make meaningful inferences about the conserved developmental functions of this family that have helped mold animal body plans.

## Results

### Phylogenetic reconstruction of *Nr2f* evolution in animals

Although previous work has investigated the homology of some Nr2fs within metazoans, these analyses were primarily focused on their relationship to other NRs and were limited by the comparatively little genomic information at the time [1, 3, 14, 43, 44]. Therefore, the relatively few Nr2f family members examined in the previous analysis did not provide a specific and detailed understanding of Nr2f evolution. To garner a better understanding of how the *Nr2f* family has evolved in animals, we performed a phylogenetic analysis using 153 Nr2f proteins with representatives from placozoans to mammals (Fig 2; S1 File). Early-branching metazoan models *Amphimedon queenslandica* (sponge) and *Mnemiopsis leidyi* (ctenophore) were not included, as we did not find putative *Nr2f* orthologs based on current databases, consistent with published phylogenetic studies of the NR superfamily [13, 15]. The placozoan *Trichoplax adhaerens* Nr2f, which was previously shown to cluster with vertebrate Nr2fs in phylogenetic analyses [45], was used as the outgroup in a maximum-likelihood (ML) phylogenetic tree. Protein sequences from groups that caused long branch artifacts due to significant divergence were not included in the phylogenetic trees (S2 File). This phylogenetic analysis provided

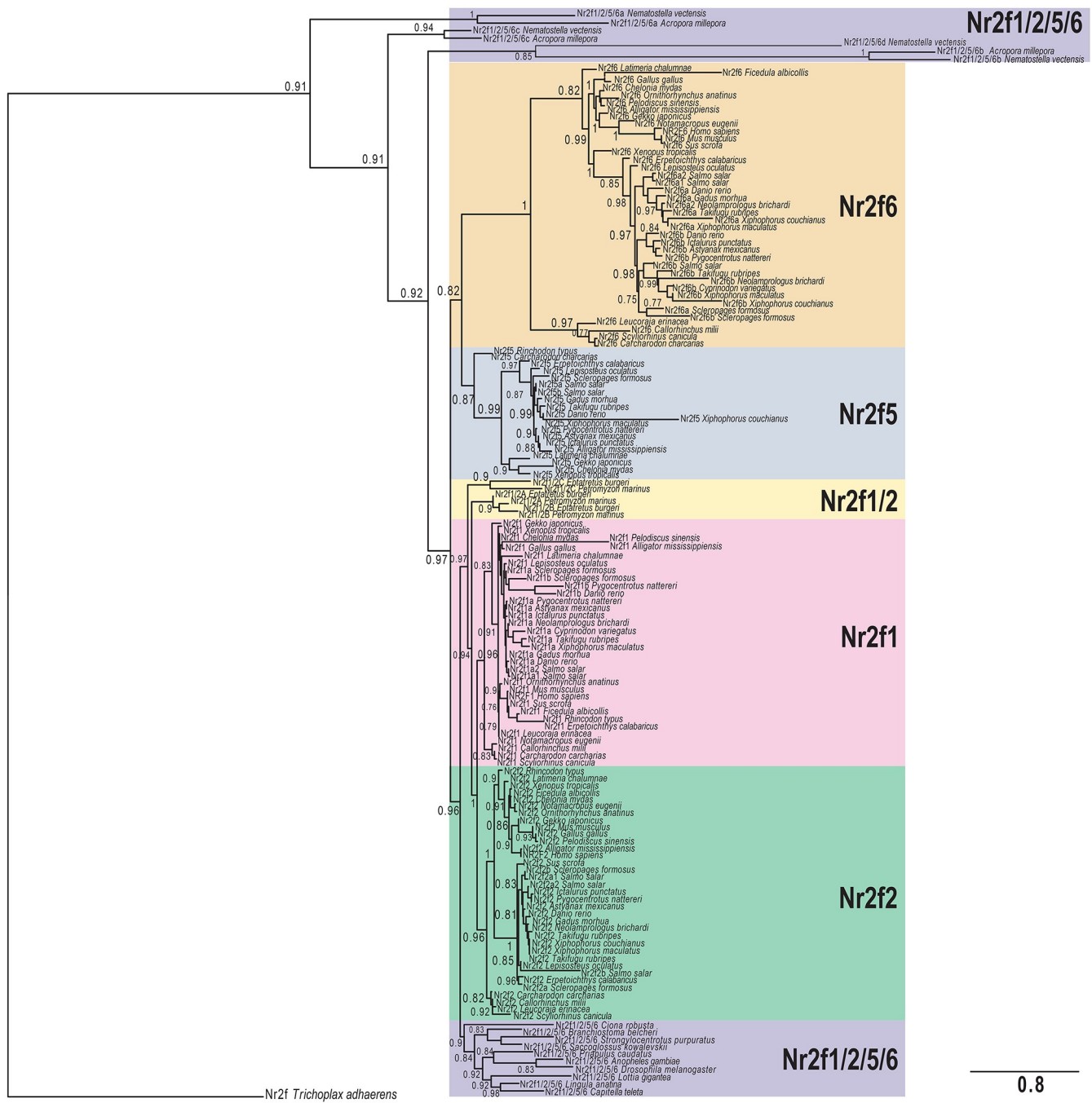

**Fig 2. Evolutionary reconstruction of Nr2f TFs in metazoans.** Phylogenetic tree of Nr2f members demonstrate the existence of six protein classes: Nr2f1/2/5/6 (violet box), Nr2f1/2 (yellow box), Nr2f1 (pink box), Nr2f2 (green box), Nr2f5 (blue box), Nr2f6 (orange box). The placozoan *Trichoplax adhaerens* Nr2f was used as the outgroup. Values at the branches indicate replicates obtained using the aLRT method.

evidence for the existence of distinct *Nr2f* subfamilies (Fig 2). Moreover, the same relationships were also supported using a Bayesian model selection (S1 Fig). Present information allowed us to identify four Nr2fs in the cnidaria *Nematostella vectensis* and *Acropora millepora*, three in *Hydra vulgaris*, and one for *Hydractinia echinata*. However, while identifiable as Nr2fs, an *A. millepora*, the *H. vulgaris*, and the *H. echinata* Nr2fs caused long-branch artifacts and were

consequently excluded (S2 File). Interestingly, the tree incorporating the *N. vectensis* and remaining *A. millepora* Nr2f members, which we now call Nr2f1/2/5/6a-d based on their relationship to Bilateria Nr2fs, were found at the base of the eumetazoan Nr2f proteins and are likely the result of gene duplications within cnidaria [15] (Fig 2). The protostome and deuterostome *Nr2f* sequences clustered into six subfamilies, which we have called *Nr2f1/2/5/6*, *Nr2f1/2*, *Nr2f1*, *Nr2f2*, *Nr2f5*, and *Nr2f6*. Single *Nr2f1/2/5/6* subfamily genes, which are highly conserved, yet evolutionary divergent from the *Nr2f1/2/5/6* genes present in early-branching eumetazoa, were found in invertebrate protostomes, invertebrate deuterostomes (hemichordates, echinoderms), and invertebrate chordates (amphioxus, tunicates). An older nomenclature proposal suggested that the *Drosophila* Nr2f (*svp*) should be designated *Nr2f3* [19], implying the other invertebrate *Nr2fs* should follow this nomenclature. However, this designation seems to obfuscate the homology of these genes revealed here and imply a different evolutionary relationship, as there is no distinct *Nr2f3* subfamily. Thus, we propose using *Nr2f1/2/5/6* or in the future potentially just *Nr2f* for the early-branching eumetazoan and invertebrate *Nr2f* genes. We have used *Nr2f1/2/5/6* in this manuscript to refer to the invertebrate *Nr2fs* to reinforce their evolutionary relationship within the Nr2f family. The invertebrate Nr2f1/2/5/6 group is more closely related to the branch that includes Nr2f1/2s from the agnathan (lamprey and hagfish) and vertebrate Nr2f1 and Nr2f2 proteins than the vertebrate Nr2f5 and Nr2f6 subfamilies (Fig 2). The clustering of the invertebrate Nr2f1/2/5/6 and agnathan Nr2f1/2 proteins with Nr2f1 and Nr2f2 of gnathostomes suggests that these paralogous gnathostome genes arose from distinct duplicative events during vertebrate evolution [41, 42]. In addition, the three agnathan Nr2f proteins found in Sea lamprey (*Petromyzon marinus*) and hagfish (*Eptatretus burgeri*) (Fig 2), which we have called Nr2f1/2A, Nr2f1/2B, and Nr2f1/2C, diverge and cluster together at the base of the vertebrate Nr2f1 and Nr2f2 proteins (Fig 2), supporting that the duplications leading to these proteins in agnathans were distinct from those that gave rise to the Nr2f paralogs in gnathostomes.

Our analysis also shows that Nr2f5 and Nr2f6 form a separate branch and are sisters groups, implying that they are paralogous and derived from the second of the vertebrate WGDs [41, 42]. Importantly, while all gnathostomes examined have retained Nr2f1, Nr2f2 and Nr2f6, current genomic data support that Nr2f5 has been independently lost by multiple vertebrate groups. Cartilaginous fish, including Whale shark (*Rinchodon typus*) [46] and the Great white shark (*Carcharadon charcarias*) [47], have retained *Nr2f5* genes, while they are absent in chimaera [48] and skates (S3 File). In amniotes, *Nr2f5* genes were found in reptiles, such as American alligator (*Alligator mississippiensis*), gecko (*Gekko japonicus*), and the Green sea turtle (*Chelonia mydas*) (Fig 2; S1 Fig), but absent from the Chinese sea turtle (*Pelodiscus sinesis*), as well as birds and mammals (S3 File). Although previous work had also designated a *Xenopus laevis Nr2f4* [19], our data indicate there is no evidence for a separate *Nr2f4* subfamily and that this gene should be called *Nr2f5*. Comparing the vertebrate Nr2f1/Nr2f2 and Nr2f5/Nr2f6 clusters, the branching and distances from our phylogenetic trees indicate that Nr2f1 and Nr2f2 are more highly conserved, while Nr2f6 TFs comprise the most divergent vertebrate Nr2f subfamily (Fig 2; S1 Fig).

To analyze the impact of additional WGDs on *Nr2f* genes, which took place in teleosts [17, 18], and specifically, in salmonids [49], we surveyed the Nr2f proteins of 12 teleost species (Fig 2; S1 Fig). Consistent with the WGDs in these species, there was a tremendous expansion of the *Nr2f* family in this clade, although it was accompanied by differential *Nr2f* paralog losses in some species (Fig 2; S1 Fig). To further interrogate the evolution of the Nr2f proteins, we examined alignments of the highly conserved zinc-fingers (Znf) within their DNA-binding domains (DBDs) using representatives from each subfamily (Fig 3). Although there is a high degree of conservation in all the examined Nr2fs, the amino acid changes in the DBDs parallels

**Znf I**      **Znf II**

```
T. adhaerens      Nr2f          CLICGDRSNGRHYGVISCEGC  CTCSANCKITKANRNQCQFC
N. vectensis      Nr2f1/2/5/6a  CAVCGDKSSGKHYGVFTCEGC  CRASRDCPIDQHHRNQCQYC
A. millepora      Nr2f1/2/5/6a  CAVCGDKSSGKHYGVFTCEGC  CRASRNCPIDQHHRNQCQYC
N. vectensis      Nr2f1/2/5/6b  CAVCGDKSTGKHYGVSTCEGC  CRGQNTCAIDRNSRSRCPSC
A. millepora      Nr2f1/2/5/6b  CAVCGDKSSGKHYGVFTCEGC  CRGQNTCAIDRNSRSRCPSC
N. vectensis      Nr2f1/2/5/6c  CAVCGDKSSGKHYGVYTCEGC  CRGFKNCPVDIQHRNHCQYC
A. millepora      Nr2f1/2/5/6c  CAVCGDKSTGKHYGVSTCEGC  CRGFKNCPVDIQHRNHCQYC
N. vectensis      Nr2f1/2/5/6d  CAVCSAPSSGRHYGVFTCEGC  CEGSGSCRVDKQNRTQCQAC
A. millepora      Nr2f1/2/5/6d  CSVCGDHSTGRHYGANTCEGC  CRVTGCCPVNKRYRNSCQYC

D. melanogaster   Nr2f1/2/5/6   CVVCGDKSSGKHYGQFTCEGC  CRGSRNCPIDQHHRNQCQYC
C. elegans        Nr2f1/2/5/6   CVVCGDKSSGKHYGQFSCEGC  CRATKNCAIDVQHRNQCQYC
L. gigantea       Nr2f1/2/5/6   CVVCGDKSSGKHYGQFTCEGC  CRGNKNCPIDQHHRNQCQYC
S. kowalevskii    Nr2f1/2/5/6   CVVCGDKSSGKHYGQFTCEGC  CRANRNCPIDQHHRNQCQYC
S. purpuratus     Nr2f1/2/5/6   CVVCHDKSSGKHYGQFTCEGC  CRANRNCPIDQHHRNQCQYC
C. robusta        Nr2f1/2/5/6   CVVCGDKSSGKHYGQYTCEGC  CRGNRNCPIDQHHRNQCQYC
O. dioica         Nr2f1/2/5/6   CVVCGDKSSGKHYGQFTCEGC  CRGNRSCPIDQHHRNQCQYC
B. belcheri       Nr2f1/2/5/6   CVVCGDKSSGKHYGQFTCEGC  CRGNRTCPIDQHHRNQCQYC

P. marinus        Nr2f1/2A      CVVCGDKSSGKHYGQFTCEGC  CRANRNCPIDQHHRNQCQYC
P. marinus        Nr2f1/2B      CVVCGDKSSGKHYGQFTCEGC  CRANRNCPIDQHHRNQCQYC
P. marinus        Nr2f1/2C      CVVCGDKSSGKHYGQLTCEGC  CRAARACPIDQHHRNQCQYC
E. burgeri        Nr2f1/2A      CVVCGDKSSGKHYGQFTCEGC  CRANRNCPIDQHHRNQCQYC
E. burgeri        Nr2f1/2B      CVVCGDKSSGKHYGQFTCEGC  CRANRNCPIDQHHRNQCQYC
E. burgeri        Nr2f1/2C      CVVCGDKSSGKHYGQFTCEGC  CRAARNCPIDQHHRNQCQYC

C. milii          Nr2f1         CVVCGDKSSGKHYGQFTCEGC  CRANRNCPIDQHHRNQCQYC
C. carcharias     Nr2f1         CVVCGDKSSGKHYGQFTCEGC  CRANRNCPIDQHHRNQCQYC
L. oculatus       Nr2f1         CVVCGDKSSGKHYGQFTCEGC  CRANRNCPIDQHHRNQCQYC
D. rerio          Nr2f1a        CVVCGDKSSGKHYGQFTCEGC  CRANRNCPIDQHHRNQCQYC
D. rerio          Nr2f1b        CVVCGDKSSGKHYGQFTCEGC  CRANRNCPVDQHHRNQCQYC
L. chalumnae      Nr2f1         CVVCGDKSSGKHYGQFTCEGC  CRANRNCPIDQHHRNQCQYC
X. tropicalis     Nr2f1         CVVCGDKSSGKHYGQFTCEGC  CRANRNCPIDQHHRNQCQYC
F. albicollis     Nr2f1         CVVCGDKSSGKHYGQFTCEGC  CRANRNCPIDQHHRNQCQYC
G. japonicus      Nr2f1         CVVCGDKSSGKHYGQFTCEGC  CRANRNCPIDQHHRNQCQYC
C. mydas          Nr2f1         CVVCGDKSSGKHYGQFTCEGC  CRANRNCPIDQHHRNQCQYC
H. sapiens        NR2F1         CVVCGDKSSGKHYGQFTCEGC  CRANRNCPIDQHHRNQCQYC

C. milii          Nr2f2         CVVCGDKSSGKHYGQFTCEGC  CRANRNCPIDQHHRNQCQYC
C. carcharias     Nr2f2         CVVCGDKSSGKHYGQFTCEGC  CRANRNCPIDQHHRNQCQYC
L. oculatus       Nr2f2         CVVCGDKSSGKHYGQFTCEGC  CRANRNCPIDQHHRNQCQYC
D. rerio          Nr2f2         CVVCGDKSSGKHYGQFTCEGC  CRANRNCPIDQHHRNQCQYC
L. chalumnae      Nr2f2         CVVCGDKSSGKHYGQFTCEGC  CRANRNCPIDQHHRNQCQYC
X. tropicalis     Nr2f2         CVVCGDKSSGKHYGQFTCEGC  CRANRNCPIDQHHRNQCQYC
F. albicollis     Nr2f2         CVVCGDKSSGKHYGQFTCEGC  CRANRNCPIDQHHRNQCQYC
G. japonicus      Nr2f2         CVVCGDKSSGKHYGQFTCEGC  CRANRNCPIDQHHRNQCQYC
C. mydas          Nr2f2         CVVCGDKSSGKHYGQFTCEGC  CRANRNCPIDQHHRNQCQYC
H. sapiens        NR2F2         CVVCGDKSSGKHYGQFTCEGC  CRANRNCPIDQHHRNQCQYC

C. carcharias     Nr2f5         CMVCGDKSSGKHYGQFTCEGC  CRGSRNCPIDQHHRNECQHC
L. oculatus       Nr2f5         CMVCGDKSSGKHYGQFTCEGC  CRGNRDCPIDQHHRNQCQYC
D. rerio          Nr2f5         CMVCGDKSSGKHYGQFTCEGC  CRGNRDCPIDQHHRNQCQYC
L. chalumnae      Nr2f5         CMVCGDKSSGKHYGQFTCEGC  CRGNRDCPIDQHHRNQCQYC
X. tropicalis     Nr2f5         CLVCGDKSSGKHYGQFTCEGC  CRGNRDCPIDQHHRNQCQYC
C. mydas          Nr2f5         CMVCGDKSSGKHYGQFTCEGC  CRSNRECPIDQHHRNQCQHC

C. milii          Nr2f6         CVVCGDRASGKHYGQFTCEGC  CRSNRDCHIDQHHRNQCQYC
C. carcharias     Nr2f6         CVVCGDRASGKHYGQFTCEGC  CRSNRDCQIDQHHRNQCQYC
L. oculatus       Nr2f6         CVVCGDKSSGKHYGVFTCEGC  CRSNRDCQIDQHHRNQCQYC
D. rerio          Nr2f6a        CVVCGDKSSGKHYGVFTCEGC  CRSNRDCQIDQHHRNQCQYC
D. rerio          Nr2f6b        CVVCGDKSSGKHYGVFTCEGC  CRSNRECQIDQHHRNQCQYC
L. chalumnae      Nr2f6         CVVCGDKSSGKHYGAFTCEGC  CRSNRDCQIDQHHRNQCQYC
X. tropicalis     Nr2f6         CVVCGDKSSGKHYGVFTCEGC  CRANRNCPIDQHHRNQCQYC
F. albicollis     Nr2f6         CVVCGDKSSGKHYGVFTCEGC  CRSNRDCQIDQHHRNQCQYC
G. japonicus      Nr2f6         CVVCGDKSSGKHYGVFTCEGC  CRSNRDCQIDQHHRNQCQYC
C. mydas          Nr2f6         CTVCGDKSSGKHYGVFTCEGC  CRSNRDCQIDQHHRNQCQYC
H. sapiens        NR2F6         CVVCGDKSSGKHYGVFTCEGC  CRSNRDCQIDQHHRNQCQYC
```

**Fig 3. Zinc finger (Znf) motifs within the DBD of the Nr2f family.** Alignments of first (I) and second (II) Znfs found in Nr2f TFs. Yellow represents highly conserved amino acids throughout all species. White indicates amino acids that are not conserved. Turquoise and blue indicate amino acid changes that are conserved within Znf I of Nr2f5 and -6, respectively. The valine change found in some Nr2f6 LBDs is also found in the placozoan and cnidaria Nr2fs. Magenta and red indicate amino acid changes at the same residue that are conserved within Znf II of Nr2f5 and Nr2f6, respectively. A glycine residue is also found at the same position in some cnidaria and invertebrate Nr2fs. Green indicates a conserved change found in most Nr2f5 and Nr2f6 Znf IIs.

the phylogenetic results of the whole proteins. The Nr2f1/2/5/6 proteins of early-branching eumetazoans showed a high degree of variability and multiple differences with respect to Nr2f1/2/5/6 DBDs of protostome and deuterostome invertebrates and the Nr2f DBDs in vertebrates. There is high similarity between Nr2f1/2, Nr2f1 and Nr2f2 DBDs in agnathans and gnathostomes, whereas Nr2f5 and Nr2f6 DBDs of gnathostomes exhibited specific changes that are consistent with their positions in the phylogenetic trees (Figs 2 and 3). Interestingly, single amino acid changes found in most Nr2f5 and Nr2f6 proteins are also found in some early-branching eumetazoans and invertebrate Nr2fs. However, the functional significance of these changes, if any, is not clear. Thus, our phylogenetic reconstruction of *Nr2f* genes in metazoans overall shows the presence of single orthologs in invertebrates and a significant expansion of the family in vertebrates that is punctuated with independent losses of *Nr2f5* in some cartilaginous fishes and amniotes.

### *Nr2f* genes have conserved intron codes

To complement the phylogenetic analysis of *Nr2f* genes, we first analyzed the conservation of *Nr2f* intron/exon structure [50–52]. Intron/exon junctions from early-branching eumetazoans and vertebrates matching the transcripts and the translated proteins were mapped and given a score for the intron phases (S4 File), with 0, 1 and 2 introns falling before the first, second and third bases of a codon, respectively. The introns were then mapped on a protein alignment comprising the highly conserved Nr2f protein DBDs and LBDs (S4 File). We found that two "phase 1" introns (one within the 3' end of the DBD and one within the LBD) are preserved in all the extant *Nr2f* subfamilies (Fig 4A; S4 File). However, *Nr2f6* genes also have a "phase 2" intron inside the second zinc-finger domain belonging to the DBD (Fig 4A; S4 File). The conservation of intron/exon junctions in the examined *Nr2f* genes allows two groups to be distinguished: one constituted by *Nr2f*, *Nr2f1/2/5/6*, *Nr2f1/2*, *Nr2f1*, *Nr2f2*, *Nr2f5*, and one comprising only vertebrate *Nr2f6* (Fig 4B), implying this unique intron/exon boundary originated after the duplication event that generated *Nr2f5* and *Nr2f6*. Thus, our analysis of intron/exon boundaries demonstrates the existence of a highly conserved intron code throughout eumetazoan *Nr2f* family members and the divergence of *Nr2f6* genes following the second WGD.

### Synteny analysis defines differential duplications and losses in the *Nr2f* family

In order to confirm the specific homologies indicated from the phylogenetic analysis, we next carried out an examination of synteny within the *Nr2f* genomic environments. With respect to representatives of the more ancient *Nr2f* genes, we did not find evidence of synteny between the single *Nr2f* in the placozoan *T. adhaerens* and the multiple *Nr2fs* in cnidarians. However, the location of the four *Nr2f* genes in *N. vectensis* and *A. millepora* genomes indicates they were likely derived from an initial duplication event followed by a tandem duplication event (Fig 5). Interestingly, *Mef2* and *Rbm8* homologs were associated with *Nr2f1/2/5/6b* in *N. vectensis* and *A. millepora*, while an *Arrdc* homolog is associated with *Nr2f1/2/5/6a* in *N. vectensis*. In vertebrates, *Mef2* paralogs (*Mef2c*, *Mef2b*, *Mef2b*) are associated with *Nr2f1*, *Nr2f2*, and *Nr2f6*, *Arrdc* paralogs (*Arrdc3*, *Arrdc4*, *Txnip/Arrdc6*) are associated with *Nr2f1*, *Nr2f2*, and *Nr2f5*, and *Rbm8a* is associated with *Nr2f5* (Figs 6–8), implying an ancient association of these genes within eumetazoan genomes.

In invertebrates, despite the synteny suggested between *Nr2f*, *Mef2*, *Arrdc*, and *Rbm8* genes in cnidaria and vertebrates, we only found limited preservation of the *Nr2f1/2/5/6* loci between two slow-evolving deuterostomes: the amphioxus (*B. belcheri)* [53] and the hemichordate

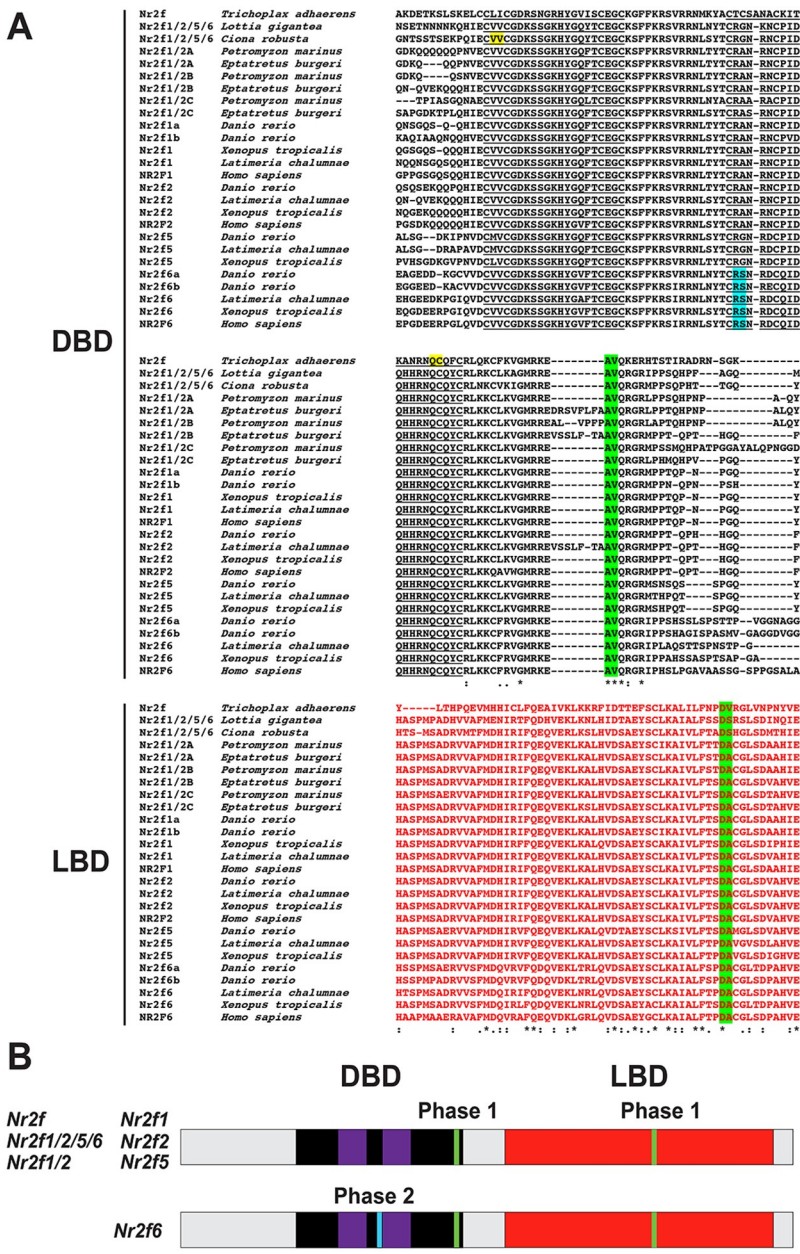

**Fig 4. Intron code of the *Nr2f* family in metazoans.** (A) Protein alignment showing conservation of intron/exon structures within the DBDs (black) and LBDs (red) of Nr2f members. Znfs in the DBDs are underlined. Phase 0 introns—yellow, phase 1 introns—green, and phase 2 introns—turquoise. Asterisks indicate 100% amino acid conservation. Colons indicate high levels (>90%) amino acid conservation. Periods indicate moderate levels (50–89%) of amino acid conservation. (B) Schematization of intron/exon boundaries of *Nr2f* genes as they relate the Nr2f protein DBD and LBDs. Black box indicates DBD. Purple boxes represent the zinc-fingers motifs within the DBD. Red boxes indicate the LDB. Colored bars indicate the conserved Nr2f Phase 1 introns (green) and the *Nr2f6*-specific Phase 2 intron (turquoise).

(*S. kowalevskii*) [54] (S2 Fig). However, the limited synteny still corroborates the existence of the invertebrate Nr2f1/2/5/6 cluster shown in the phylogenetic trees (Fig 2; S1 Fig). Furthermore, the only remaining synteny between *Nr2f1/2/5/6* in invertebrates and vertebrate orthologs appears to be the linkage between *UNC45A* and *NR2F2* of primates and *Unc45a* and

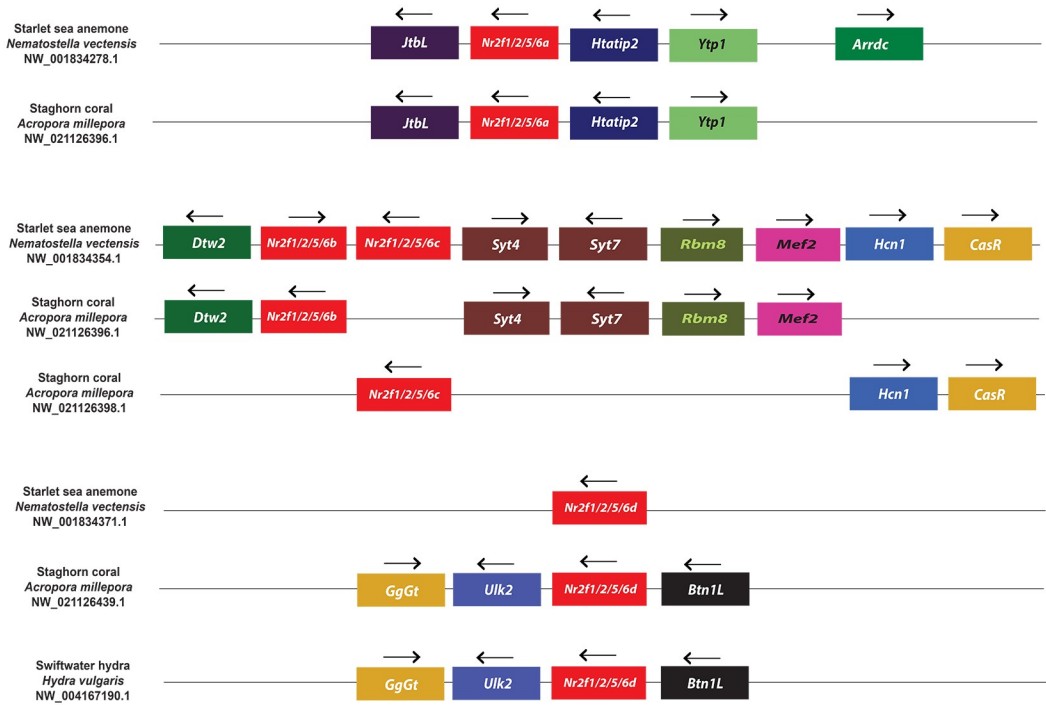

**Fig 5. Synteny analysis of *Nr2f1/2/5/6* genes in cnidaria.** Schematic of the loci flanking *Nr2f1/2/5/6* gene duplications in cnidaria *N. vectensis*, *A. millepora*, and *H vulgaris*. Only the one *H. vulgaris* gene presently has available genomic information. Arrows indicate transcription orientation.

*Nr2f1/2/5/6* of the tunicate *C. robusta* (S3 Fig), which is considered the closest living relative of vertebrates [55]. Focusing on *Nr2f1* and *Nr2f2* in the genomes of gnathostomes, including Great white sharks, coelacanths, spotted gars, zebrafish, chickens, and humans, we found a high degree of synteny for *Nr2f1* and *Nr2f2* loci and conservation of the location of flanking genes among these taxa (Fig 6). Specifically, *Nr2f1* and *Nr2f2* genes exhibited remarkably conserved syntenic environments, clustering with putative orthologs belonging to other families. *Lysmd3*, *Arrdc3*, *Mctp1* and *Mef2a* flank *Nr2f1* orthologs, while *Nr2f2* orthologs are flanked by *Lysmd4*, *Arrdc4*, *Mctp2* and *Mef2c* paralogs. Furthermore, in teleosts like zebrafish, two *Nr2f1* Ohnologs (*nr2f1a* and *nr2f1b*) also shared significant conservation of paralogous genes (Fig 6), which is consistent with an origin from the teleost-specific genome duplication (TSGD) [17, 18]. However, the *nr2f1b* gene has been lost by several teleost species (Fig 2; S1 Fig). Although the genomic information is somewhat fragmented, orthologs of flanking genes found in gnathostome *Nr2f1* and *Nr2f2*, such as *Arrdc2/3*, *Lysmd3*, *Fam172a*, were also found near each of the three Sea lamprey *Nr2f1/2* genes and the hagfish *Nr2f1/2C* gene (S4 Fig), which is consistent with these genes arising from genome duplication(s) within the agnathan lineage [56]. Together, these results suggest that *Nr2f1* and *Nr2f2* of gnathostomes have a common origin and are derived from a WGD event [41, 42].

Examining *Nr2f5* loci in representative gnathostomes showed a high degree of conservation in both species that have retained and lost the gene. The adjacent genomic environments in the majority of examined *Nr2f5* loci have retained an association with *Rbm8a* (Fig 7), whose homolog in cnidarians flanks *Nr2f1/2/5/6b* (Fig 5). The synteny is generally not shared with gnathostome *Nr2f1* and *Nr2f2* orthologs (Fig 6). However, the *Nr2f5* loci in coelacanth and amphibians have retained *Txnip* (Fig 7), which is also named *Arrdc6*. As aforementioned,

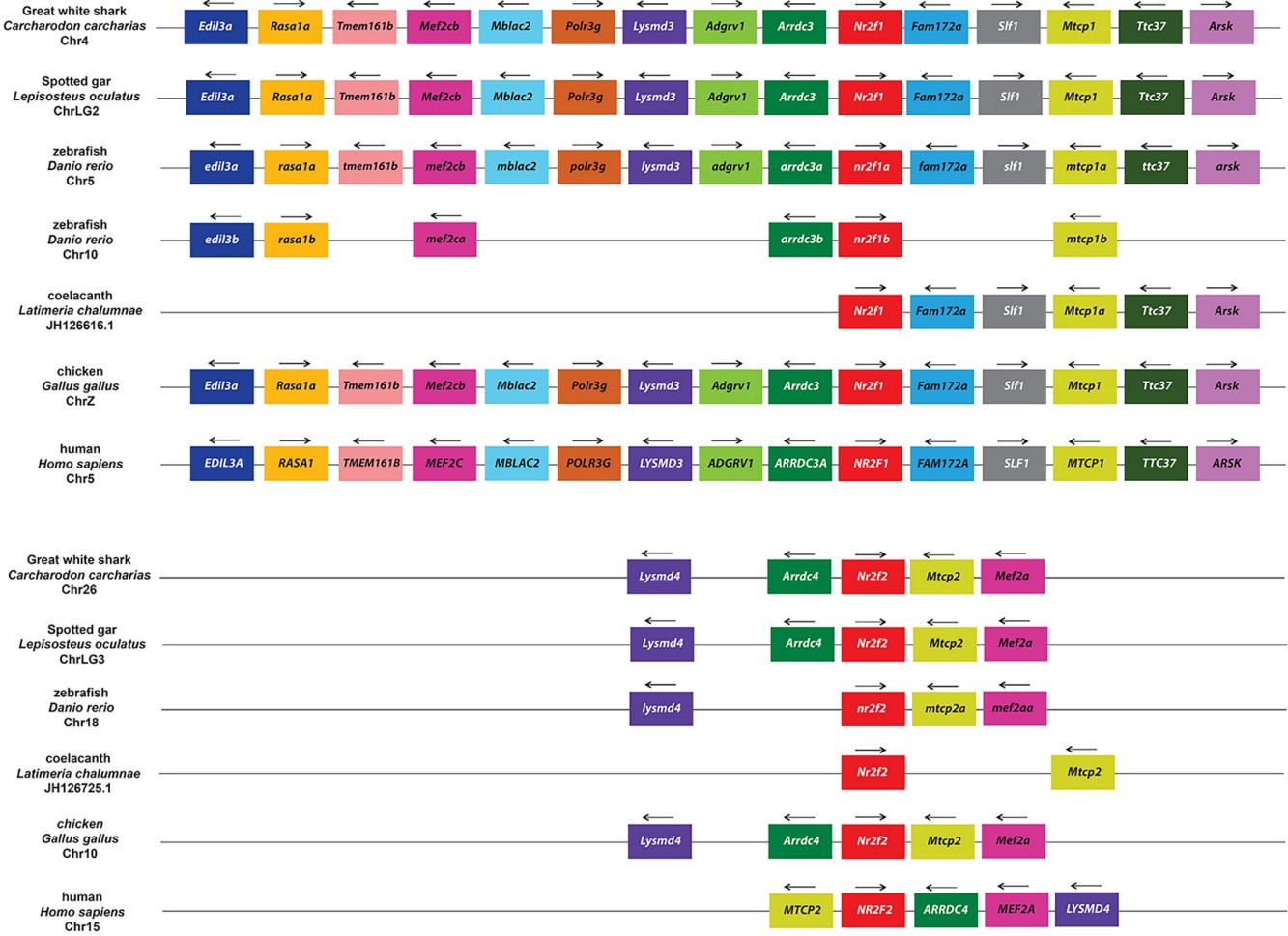

**Fig 6. Synteny analysis of vertebrate *Nr2f1* and *Nr2f2* genes.** Schematization of conserved genomic environments of gnathostome *Nr2f1* and *Nr2f2* genes (red rectangles) in selected species with relative chromosomes/scaffolds. Flanking orthologous genes are represented employing rectangles of the same color. Arrows indicate transcription orientation.

*Arrdc* family members flank the *N. vectensis Nr2f1/2/5/6a* (Fig 5) and both *Nr2f1* and *Nr2f2* genes (Fig 6). Interestingly, amniotes that have lost *Nr2f5* (representatives including Chinese soft-shell turtles, chickens, and humans) (Fig 7; S3 File) have largely preserved the flanking genomic loci that are present in cartilaginous fish, zebrafish, coelacanth, frogs, and Green see turtles (Fig 7). In contrast, the absence of *Nr2f5* in some cartilaginous fish, such as *C. milii*, correlates with the lack of the entire locus. Within the *Actinopterygii* (ray-finned fishes), the synteny of genes has been lost only on one side of the *Nr2f5* loci (Fig 7). With respect to the lamprey, its *Nr2f1/2C* ortholog is flanked by a *Bola1* ortholog, as well as orthologs of genes that flank gnathostome *Nr2f1* and *Nr2f2* (S4 Fig; Figs 6 and 7), which further suggests ancestral linkage with the single *Nr2f1/2/5/6* genes (Fig 2; S1 Fig).

As might be expected given the divergence, the *Nr2f6* subfamily did not share many common elements with the other *Nr2f* loci in gnathostomes (Fig 8). However, *Mef2b* was syntenic in Great white sharks, Spotted gar, chicken, and human genomes, similar to cnidarian *Nr2f1/2/5/6b* (Fig 5) and gnathostome *Nr2f1* and *Nr2f2* (Fig 6). Within gnathostomes, the *Nr2f6* loci were highly conserved from cartilaginous fish to mammals, although there were significant gene losses surrounding *nr2f6a* and *nr2f6b* loci in zebrafish and one side of the *Nr2f6* locus in

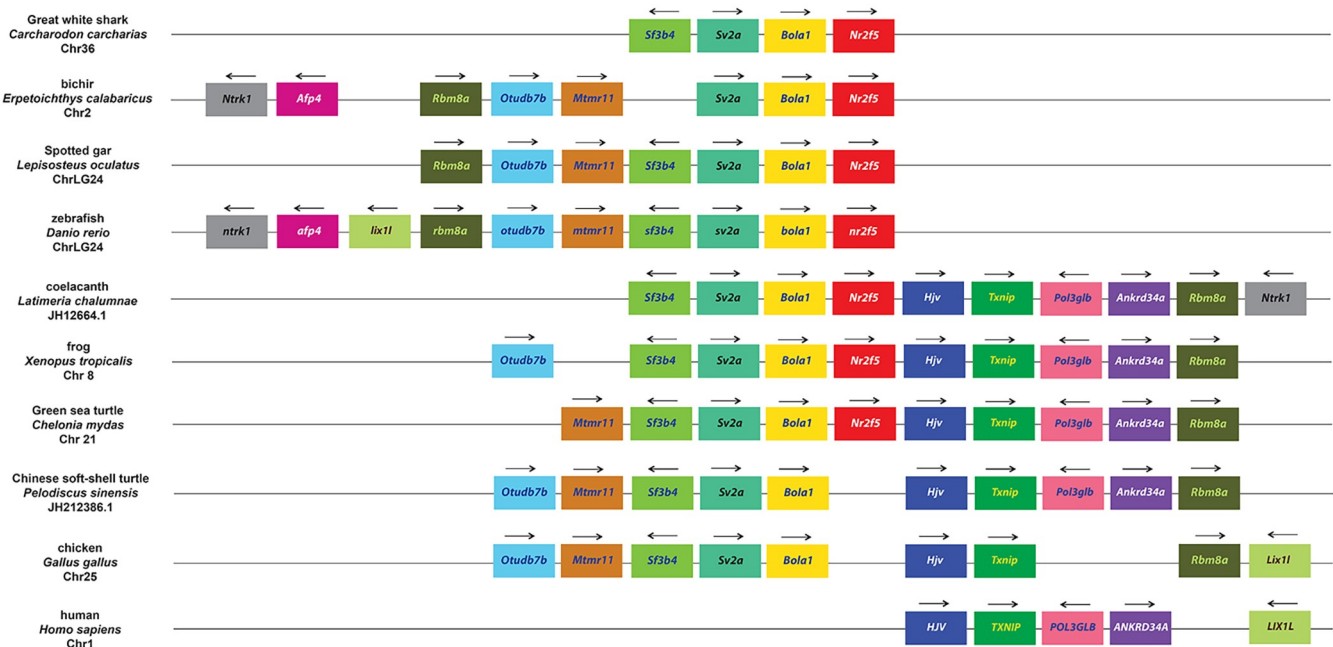

**Fig 7. Synteny analysis of vertebrate *Nr2f5* genes.** Schematization of conserved genomic environments of gnathostome *Nr2f5* genes (red rectangles) in selected species with relative chromosomes/scaffolds. Flanking orthologous genes are represented using rectangles of the same color. Arrows indicate transcription orientation.

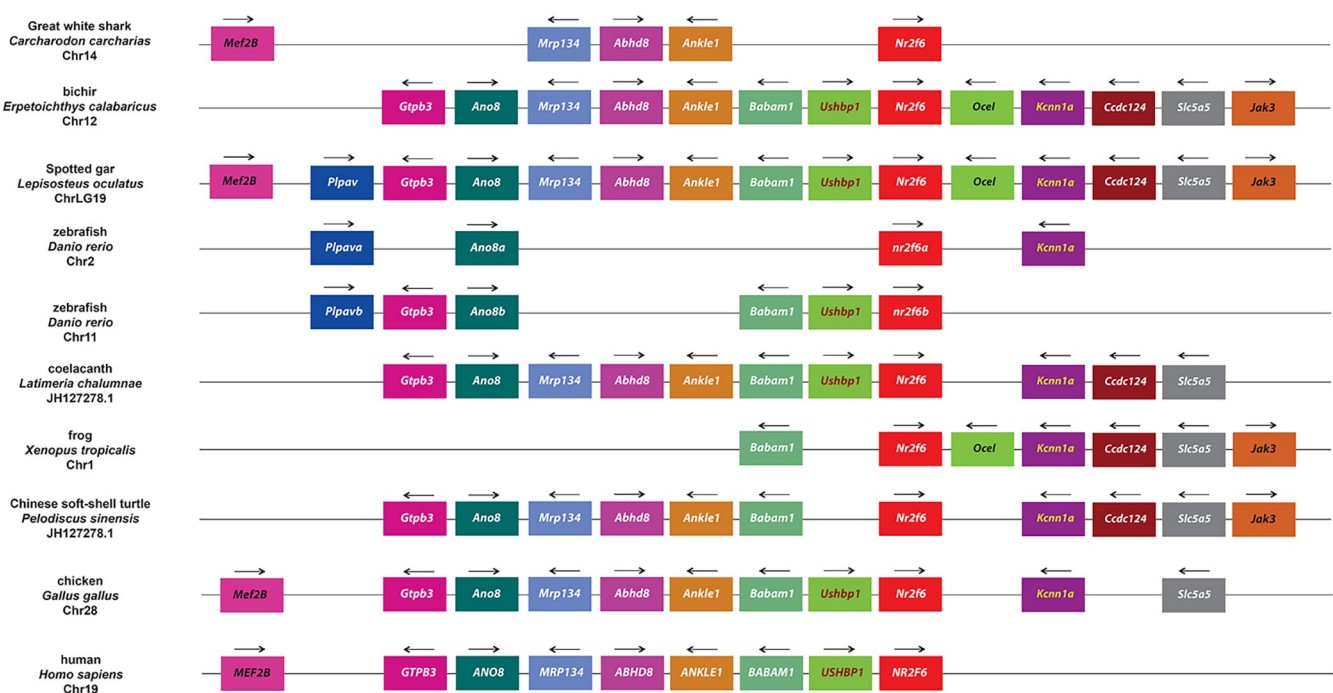

**Fig 8. Synteny analysis of vertebrate *Nr2f6* genes.** Schematization of conserved genomic environments of gnathostome *Nr2f6* genes (red rectangles) in selected species with relative chromosomes/scaffolds. Flanking orthologous genes are represented using the same color code. Arrows indicate transcription orientation.

humans. Furthermore, the presence of conserved orthologs (*ano8a* and *ano8b*, *plvapa* and *plvapb*) flanking *nr2f6a* and *nr2f6b* zebrafish genes suggested that they originated from the TSGD. Together, these findings show that despite the greater divergence of the *Nr2f5* and *Nr2f6* within vertebrates the genomic environments have retained some synteny and surrounding *Nr2f5* and *Nr2f6* loci are highly conserved within gnathostomes.

### Effects of TSGD on the *Nr2f* gene repertoire

We next wanted to measure the impact of the series of additional WGDs that have occurred in teleosts on *Nr2f* gene number (Fig 2; S1 Fig). For this comparison, we examined all the *Nr2f* loci in zebrafish, the Asian arowana (*S. formousus*), which is documented to retain duplicates [57], and the Atlantic salmon (*S. salar*), which has a salmonid-specific genome duplication (SSGD) [49]. We found that each of these teleosts retained two *Nr2f1* Ohnologs (Fig 9), suggesting they either were not duplicated or that one pair of Ohnologs was lost in salmonids. Zebrafish lost one *nr2f2* Ohnolog, maintaining only the *nr2f2a* ortholog, while Asian arowana retained two *Nr2f2* Ohnologs. Salmonids have 3 *Nr2f2* genes (*Nr2f2a1*, *Nr2f2a2*, and *Nr2f2b1*), due to a loss of the one of the *Nr2f2b* Ohnologs following their additional WGD. With respect to *Nr2f5*, only Atlantic salmon showed two copies, implying these were generated during the SSGD event, as suggested by the presence of two *Nr2f5* Ohnologs in other salmonids (*Oncorhynchus spp.*, *Coregonus clupeaformis*) (S5 File). Finally, zebrafish and Asian arowana each possess two *Nr2f6* genes, while Atlantic salmon has 3 similar to what is found in the *Nr2f2* subfamily (Fig 9). Inspecting other teleost *Nr2f* gene family repertoires (Fig 2; S1 Fig), we

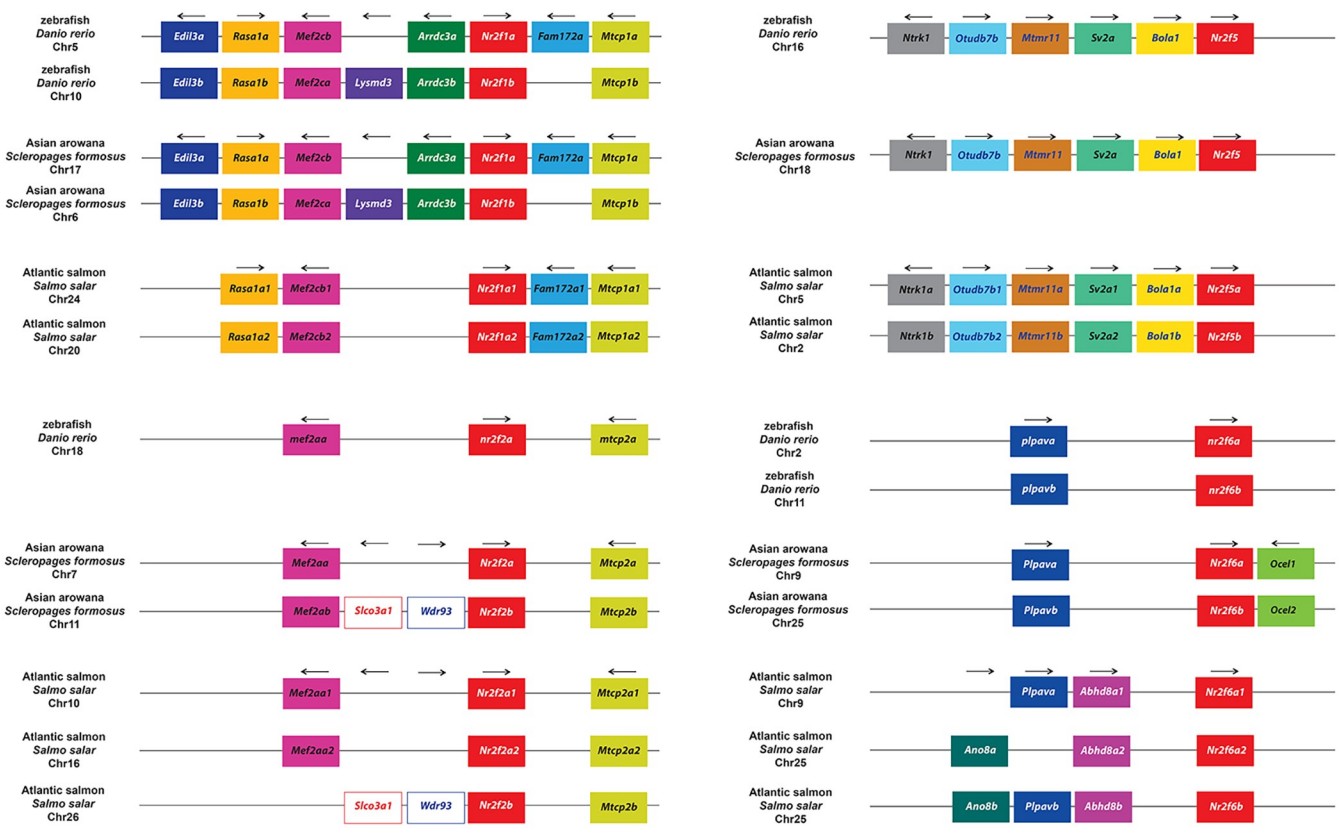

**Fig 9. Synteny analysis of *Nr2f* genes in teleosts.** Comparison of *Nr2f* genome environments in selected teleosts (zebrafish, Asian arowana, Atlantic salmon) with relative chromosomes/scaffolds. Rectangles of the same color represent flanking orthologous genes. Arrows indicate transcription orientation.

found that the Channel catfish (*Ictalurus punctatus*), Red-bellied piranha (*Pygocentrus nattereri*), cavefish (*Astyanax mexicanus*) and Sheepshead minnow (*Cyprinodon variegatus*) all retained only *Nr2f6b*. The Sheepshead minnow and Princess cichlid (*Neolamprologus brichardi*) also lost *Nr2f5*. However, other cichlids like Nile tilapia (*Oreochromis niloticus*) and Zebra mbuna (*Maylandia zebra*) did not lose *Nr2f5* (S5 File). Intriguingly, the Monterrey platifish (*Xiphophorus couchianus*) is the only gnathostome without any *Nr2f1* paralogs, differing from its sibling species, the common platifish (*X. maculatus*), which possesses *Nr2f1a*. Therefore, teleosts show an expansion of *Nr2f* genes following TSGD and SSGD, which were followed by high variability in species-specific losses of *Nr2f* Ohnologs.

## Discussion

We have performed an examination of *Nr2f* gene evolution in metazoans. Our analysis corroborates previous work showing that *Nr2f* genes are present in some representative early-branching eumetazoans (placozoans and cnidarians) [15, 58], but that they are absent in early-branching metazoans, i.e. sponges and ctenophores [15, 58]. Importantly, our data support a model in which a single *Nr2f* gene, which is present in a representative placozoan, predated a *Nr2f1/2/5/6* subfamily found in cnidaria and six Bilateria subfamilies that include *Nr2f1/2/5/6* (found in invertebrate protostome and deuterostomes), *Nr2f1/2* (found in agnathans), and *Nr2f1*, *Nr2f2*, *Nr2f5*, and *Nr2f6* (found in vertebrates; Fig 10). Single, conserved *Nr2f1/2/5/6* genes are predominantly found throughout invertebrate protostomes and deuterostomes and have even been retained in species traditionally considered gene losers, such as the tunicates [52, 59, 60]. There has been significant expansion and retention of *Nr2fs* in gnathostomes, particularly in teleosts. Although initial analysis in lampreys suggested they may possess only one *Nr2f* gene [61], our evolutionary assessment shows that extant agnathans have three *Nr2f* members, which appear to have originated in part from an agnathan WGD event [56]. Interestingly, the single Nr2f1/2/5/6 proteins in invertebrates are also highly conserved at the

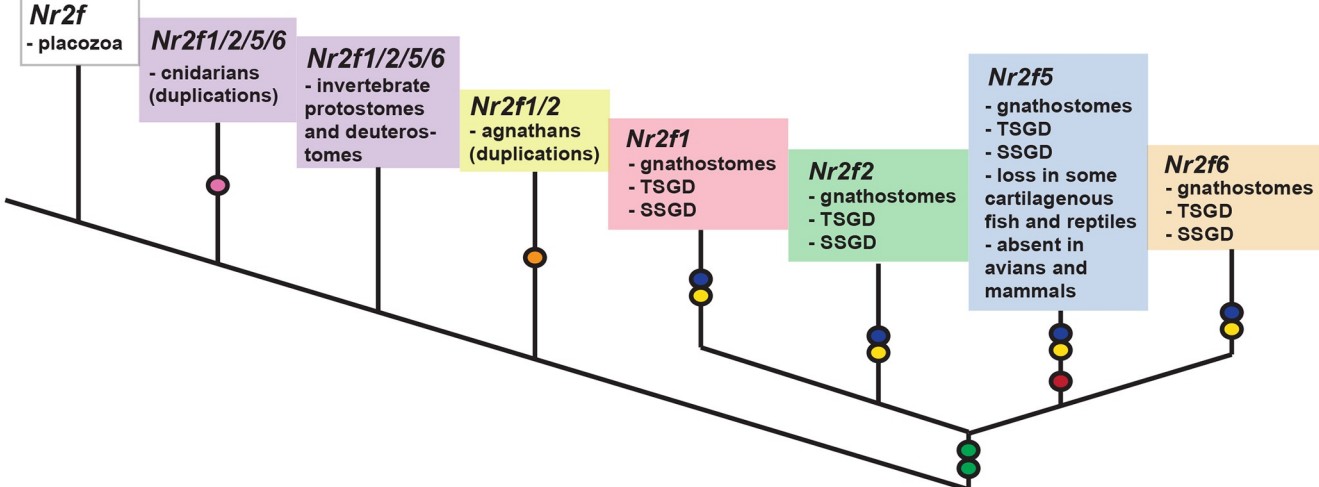

**Fig 10. Model summarizing the evolutionary events of the *Nr2f* family in Metazoa.** A single *Nr2f* of placozoans (white box) represents the ancestor of extant *Nr2fs*. There were duplicative events specific to cnidaria leading to the expansion of *Nr2f1/2/5/6* (pink circle). Invertebrate protostomes and deuterostomes have predominantly retained a single *Nr2f1/2/5/6* homolog. There were duplicative events specific to agnathans leading to an expansion of *Nr2f1/2* genes (orange circle). WGDs within vertebrates (green circles) generated the four *Nr2fs* found in vertebrates, with *Nr2f1/Nr2f2* being paralogous and *Nr2f5/Nr2f6* being paralogous. *Nr2f5* has been independently lost in multiple vertebrate groups (red circle). It is lost in some cartilaginous fish and turtles (reptilian amniotes), and is absent in avian and mammalian amniotes. Teleosts have additional *Nr2f* Ohnologs due to TSGDs (blue circles) and SSGDs (yellow circles).

sequence level and cluster with the Nr2f1/2 proteins in agnathans and Nr2f1 and Nr2f2 proteins in gnathostomes. Furthermore, our data support a parsimonious view that *Nr2f1* and *Nr2f2* are paralogous and *Nr2f5* and *Nr2f6* are paralogous, consistent with each of the Nr2f1/2 and Nr2f5/6 branches being created from an initial WGD [41, 42]. Within gnathostomes, the genomic environments of each the *Nr2f1*, *Nr2f2*, *Nr2f5*, and *Nr2f6* orthologs have retained significant synteny of their loci [16, 21, 22]. Remarkably, while limited synteny exists between the *Nr2f1/2* and *Nr2f5/6* branches and within the *Nr2f5/6* branch, members of these families have retained association with *Mef2*, *Arrdc*, and *Rbm8* homologs within their genomic environments, which is also found in cnidaria. However, this genomic association was not found in other examined invertebrate genomes. Our analysis also shows the *Nr2f5* subfamily is the smallest in vertebrates, having been independently lost in multiple gnathostomes (some cartilaginous fishes, amniotes—some reptiles, absent in birds and mammals) (Fig 10). In contrast to *Nr2f5*, the *Nr2f6* subfamily has been retained by all the evaluated gnathostomes, despite being the most divergent at the sequence level, with respect to synteny, and intron/exon structure.

Although overall there has been relatively limited comparative analysis of *Nr2f* gene expression beyond major model organisms, integrating our phylogenetic assessment with available expression and functional analyses of the *Nr2f* members in evolutionarily distant animals [12, 14] presently supports a hypothesis that Nr2f expression originated in neural tissue and regulation of neuronal differentiation may be the most ancient *Nr2f* function. Foremost, the two *Nr2f* members (both *Nr2f1/2/5/6c*) of the diploblastic cnidaria *H. vulgaris* and *H. echinata* thus far examined appear to be expressed in neurons and have requirements in neurogenesis [12, 14]. Clearly, the expression of the additional *Nr2f* cnidarian homologs that have been identified needs to be examined and if found to be expressed in endoderm would alter this hypothesis. Nevertheless, the function of *Nr2f1/2/5/6* orthologs of protostome invertebrates nematodes and flies have been extensively studied in neural tissues and neural sensory cell differentiation [8, 62, 63]. In invertebrate deuterostomes, the single *Nr2f1/2/5/6* orthologs are expressed in neural tissue of sea urchin (*Strongylocentrotus purpuratus*), amphioxus, and sea squirt embryos [11, 64–66]. Recent functional analysis of the Mediterranean sea urchin (*Paracentrotus lividus*) Nr2f1/2/5/6 shows that it is required for the development of neural and ectodermal derivatives [67]. A *Nr2f1/2/5/6* ortholog from the agnathan River lamprey (*Lampetra fluviatilis*) is also expressed in the developing nervous system [61]. However, our identification of three *Nr2f1/2* members in agnathans suggests that additional expression and potentially functional analysis should be performed in the Sea lamprey (*P. marinus*) and/or hagfish (*E. burgeri*) to understand the conservation of the different agnathan paralogs compared to Nr2fs in vertebrates. Both *Nr2f1* and *Nr2f2* orthologs share overlapping central nervous system (CNS) expression in mouse and zebrafish [16, 21, 68]. However, *nr2f1a* and *nr2f2* are both expressed more extensively in neural tissue of zebrafish embryos, while *Nr2f1* is predominantly expressed in neural tissues of mice [21, 22]. *Nr2f5* is expressed in neural tissue and derivatives, including in the eyes of zebrafish and newts [68–70]. *Nr2f6* genes have conserved expression within the central nervous system of mammals [16], as well as both zebrafish *nr2f6* Ohnologs. Thus, all *Nr2fs* examined are expressed in neural tissue, with experiments in cnidaria and invertebrates presently supporting their ancestral requirements may be in neural cell differentiation.

While we propose that *Nr2fs* may have originated with requirements in neural differentiation, they are also required for the development of mesodermal and endodermal-derived tissues through Bilateria. Thus, it is interesting to consider some of these requirements in light of our phylogenetic analysis. In addition to neural differentiation, *Nr2f* homologs are necessary for copulation control in nematodes [71] and heart vessel specification in flies [23, 24]. Furthermore, the recent work with the Mediterranean sea urchin suggests that it is required for

the development of mesendodermal derivatives [67]. The functions of *Nr2f1* and *Nr2f2* genes have been intensely investigated in vertebrate models and they are required for proper human development [22, 31, 72]. Both expression and functional analysis of *Nr2f1* and *Nr2f2* genes in vertebrates show that they have acquired distinct developmental roles during evolution. Following overlapping expression early in mouse embryos, murine *Nr2f1* and *Nr2f2* become predominantly expressed in neural and mesendodermal tissues, respectively [21, 22]. Analysis of these *Nr2f* genes in mice and zebrafish support the functional divergence of these proteins. Murine *Nr2f1* KOs have glial differentiation defects [73], while *Nr2f2* is required for proper development of many mesendodermal-derived tissues, including atrial chamber and arterial-venous differentiation [40, 74]. Intriguingly, mouse *Nr2f2* and zebrafish *nr2f1a* are functional homologs with respect to heart development, as both are required for atrial differentiation [75], further supporting the common evolutionary origins of these paralogs. While zebrafish *nr2f2* is not required for early atrial or vein development [29], NR2F1 and NR2F2 TFs do appear to have redundant requirements, for instance promoting atrial cardiomyocyte differentiation in human embryonic stem cells [76, 77]. It is interesting that the single *Nr2f1/2/5/6* (*svp/Nr2f3*) homolog of flies is also required for dorsal vessel (heart) development [23]. However, if these similar roles in mesodermally-derived heart tissues reflect homologous requirements within Bilateria for cardiac differentiation requires functional studies from many additional model organisms [67]. With respect to analysis of the expansion of *Nr2f1* and *Nr2f2* Ohnologs in teleosts, *Nr2f1b* actually has been lost in the majority of surveyed teleosts. *Nr2f1b* zebrafish mutants are viable [78] and surprisingly do not exhibit redundancy with *nr2f1a* in atrial cardiomyocyte differentiation [29], but do exhibit some redundancy with multiple other *Nr2f* genes in neural crest cells that promote jaw development [78]. Virtually all the analyzed gnathostome genomes have a single *Nr2f2* gene, excluding the teleosts *S. formosus* (2) and *S. salar* (3), implying there may be some dosage sensitivity that favors the retention of single orthologs in gnathostomes.

With respect to the function of *Nr2f5* and *Nr2f6* genes, zebrafish *nr2f5* mutants are viable, yet like zebrafish *nr2f1b* mutants they function redundantly with other *nr2f* genes for proper upper-jaw development [78]. While expression and functional analysis from other organisms that have retained *Nr2f5* (coelacanth, spotted gar, and frog) may provide insights into conservation of *Nr2f5* orthologs, the independent loss of *Nr2f5* genes in multiple vertebrate lineages, as well as the lack of overt requirements alone in zebrafish, suggests that *Nr2f5* orthologs likely have retained minimal developmental requirements and its loss can be tolerated. Murine *Nr2f6* KO mice have forebrain defects. Specifically, these mutants show a loss of neurons that regulate the circadian clock genes [79]. However, *Nr2f6* also has a critical role in lymphocyte differentiation and T-cell mediated tumor surveillance, suggesting requirements in mesodermally-derived tissues and neofunctionalization in adaptive immunity [80, 81]. Altogether, minimally, expression and functional data support requirements for Nr2fs in all three germ layers of Bilateria. However, the conservation of these requirements and if they reflect homologous roles in the different germ layers throughout Bilateria is not yet as clear.

In examining the evolution of the Nr2f TFs, it is also worthwhile to note that in early-branching eumetazoans through invertebrate chordates and gnathostomes there is conserved responsiveness to retinoic acid (RA) signaling [82], a critical molecule involved early patterning of vertebrate embryos [83–85], implying this relationship may form the core of an ancient gene regulatory network. *Nr2f* genes from placozoans [58] and the invertebrate chordates *Ciona* and amphioxus are all RA-responsive [11, 64]. Furthermore, in vertebrates, where the earliest requirement for RA is posteriorization of the embryo [86], virtually all the *Nr2f* genes have been shown to be responsive to RA signaling in developmental contexts involving all three germ layers. Specifically, RA signaling has been shown to positively regulate all the *Nr2fs*

in zebrafish in the developing zebrafish endoderm [87], the CNS [68], and anterior lateral plate mesoderm (ALPM) [29]. RA signaling also positively regulates *Nr2f1*, *Nr2f2* and *Nr2f6* in mice [88, 89], and *NR2F1* and *NR2F2* in humans [90, 91]. Nr2fs can inhibit RA signaling in some contexts, suggesting it may form a negative feedback loop. One role Nr2fs may play is through direct competition with retinoic acid receptors (RARs) in binding retinoic acid response elements (RAREs) [21]. Moreover, it has been shown that the cnidarian Nr2f1/2/5/6c possesses the ability to inhibit RA signaling in *in vitro* signaling assays [14]. Thus, the responsiveness of the *Nr2f* family to RA may have evolved very early and has been highly maintained through the diversification of multiple vertebrate *Nr2f* genes, implying there is high selection to maintain this relationship.

## Conclusions

Overall, our evolutionary assessment sheds new light on the events that have shaped the extant *Nr2f* family in Metazoa. The phylogenetic analysis defines the individual *Nr2f* subfamilies and their relationships across metazoan phyla, which complements available expression and functional data presently supporting an origin of their requirements in the development of neural tissue. Interestingly, the functions of Nr2f proteins are found to regulate development of all germ layers of Bilateria. The detailed evolutionary understanding of the *Nr2f* gene family we now have will allow us to infer more meaningful conclusions about the origins and conserved requirements of *Nr2f* genes in normal metazoan development and their role in sculpting diverse body plans.

## Methods

### Ethics statement

Ethical approval is not required. No animals were used in this study.

### Genome database searches and phylogenetic reconstruction

*Homo sapiens* NR2F protein sequences were employed as queries in BLASTp and tBLASTn in genome databases of selected species (NCBI, Ensembl, Ensembl Metazoa, SkateBase [92], ANISEED [93]). The entire dataset of protein sequences for domain architecture was analyzed by using the domain database provided by Expasy, named PROSITE [94] and then, manually annotated. All the surveyed sequences were verified to be Nr2f proteins through analysis of DBDs and LBDs (S6 File). The analysis was weighted with 30 species from agnathans to primates to take into account the impact of multiple WGDs in vertebrates [41, 42] and in teleosts [17, 18]. Orthology of the *Nr2f* members was initially assessed by using a reciprocal best blast hit (RBBH) approach employing default parameters and corroborated by phylogenetic analyses. Protein alignment for phylogeny was generated using L-INS-i (accurate; for alignment of <200 sequences) on MAFFT [95, 96] (S7 File). The phylogenetic reconstruction of Fig 2 was performed on the entire protein sequences and based on maximum-likelihood (ML) inferences calculated with PhyML 3.0 [97], employing automatic Akaike Information Criterion (AIC) by Smart Model Substitution (SMS) [98], which selected the JTT+G+F model employing discrete gamma distribution in categories. All parameters (gamma shape = 0.7; proportion of invariants (fixed) = 0.000) were established from the dataset. Branch support was provided by aLRT [99]. The phylogeny of S1 Fig was carried out employing Bayesian Information Criterion (BIC) by SMS, which sorted the JTT+G+F model using discrete gamma distribution in categories. All parameters (gamma shape = 0.7; proportion of invariants (fixed) = 0.000) were established from the dataset, with branch support calculated employing aBayes method [100].

Accession numbers and protein sequences used for phylogenetic tree reconstructions are provided in S1, S6, and S7 Files, while those excluded for their divergence are listed in S2 File. Common and Latin names for species used in this study are listed in S8 File.

## Analysis of intron/exon structures and phases

Gene structures were deduced after merging the genomic sequences with ESTs when available, as previously described [50–52]. Introns were classified as phase 0, phase 1, and phase 2, according to their positions with respect to the protein-reading frame. The amino-acid residues with the conserved introns were manually mapped on a ClustalX alignment [101] of selected Nr2f proteins (S9 File).

## Evaluation of synteny

We evaluated the presence/absence of synteny examining the chromosomes on public genome databases (NCBI, Ensembl, Ensembl Metazoa, ANISEED [93]). We verified the existence of duplicates using Genomicus [102] and Vertebrate Ohnologs [103]. The window considered for the locus analyses was twenty flanking genes. Genes that were not conserved were excluded from the analysis. All the genes were represented employing colored rectangles, using the same color for all *Nr2f* genes (red).

## Supporting information

**S1 Fig. Phylogenetic tree of the Nr2f family, using Bayesian Information Criterion (BIC).** The same color code as Fig 2 is used. Values at the branches indicate replicates obtained employing the aBayes method.
(TIF)

**S2 Fig. Synteny analysis of *Nr2f1/2/5/6* genes found invertebrates.** Schematic of limited conservation for *Nr2f1/2/5/6* loci between the hemichordate *S. kowalevskii* and amphioxus *B. belcheri*. Black arrows indicate transcription orientation.
(TIF)

**S3 Fig. *Unc45–Nr2f* gene duplet preservation.** Schematic of *Unc45-Nr2f* duplet conservation in genomes of ascidians (*Ciona*) and primates. The duplet is absent in other vertebrate models, including zebrafish and mouse.
(TIF)

**S4 Fig. Synteny analysis of *Nr2f1/2* genes in agnathans.** Schematic of lamprey (*P. marinus)* *Nr2f1/2* loci with relative chromosomes and available genomic data from the hagfish (*E. burgeri*). Genomic data could only be obtained for the hagfish *Nr2f1/2C* gene. Same color code of Figs 6–8 is used. Flanking genes are in common with gnathostomes, with *Arrdc2* and *Arrdc3* (green) that form a conserved duplet with *Nr2f1/2B* and *Nr2f1/2C*. *Nr2f1/2C* is adjacent to *Fam172a* in both lamprey and hagfish. Arrows indicate transcription orientation.
(TIF)

**S1 File. List of all protein sequences employed in Nr2f phylogenetic tree with accession numbers.**
(TXT)

**S2 File. List of protein sequences excluded from Nr2f phylogenetic tree due to their high degree of divergence.**
(TXT)

**S3 File. List of examined species whose genomes lacked *Nr2f5* with their common names, Latin names, and phyla.**
(XLSX)

**S4 File. Intron/Exon structure of *Nr2f* genes in Metazoa.** Alignment of specific and conserved intron/exon boundaries within the Zinc finger motifs of DBD (underlined) and LBDs (red). The intron phases have been depicted using color code: Phase 0 (yellow), Phase 1 (green), Phase 2 (turquoise).
(DOCX)

**S5 File. Sequences of additional salmonid and cichlid Nr2f5 proteins.**
(TXT)

**S6 File. Nr2f domain architectures during metazoan evolution.** Sequences used in analysis with DBDs (yellow) and LBDs (green) domains in metazoan Nr2f proteins indicated. the Zinc-finger motifs within the DBDs are underlined.
(DOCX)

**S7 File. MAFFT alignment of protein sequences used for phylogenetic analysis of Fig 2 and S1 Fig.**
(TXT)

**S8 File. List of species used for our evolutionary analyses with their common names, Latin names, and phyla.**
(XLSX)

**S9 File. Selected Nr2f transcripts and translations used for analysis with the positions and phases of intron/exon boundaries indicated.**
(DOCX)

## Author Contributions

**Conceptualization:** Ugo Coppola, Joshua S. Waxman.

**Data curation:** Ugo Coppola.

**Formal analysis:** Ugo Coppola.

**Funding acquisition:** Joshua S. Waxman.

**Investigation:** Ugo Coppola.

**Methodology:** Ugo Coppola.

**Supervision:** Joshua S. Waxman.

**Validation:** Ugo Coppola.

**Visualization:** Joshua S. Waxman.

**Writing – original draft:** Ugo Coppola, Joshua S. Waxman.

**Writing – review & editing:** Ugo Coppola, Joshua S. Waxman.

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
