## [Decision Letter · Decision Letter 0]

16 Jul 2021

PONE-D-21-18033

Origin and evolutionary landscape of Nr2f transcription factors across Metazoa

PLOS ONE

Dear Dr. Waxman,

Thank you for submitting your manuscript to PLOS ONE. After careful consideration, we feel that it has merit but does not fully meet PLOS ONE’s publication criteria as it currently stands. Therefore, we invite you to submit a revised version of the manuscript that comprehensively addresses the points raised during the review process.

We look forward to receiving your revised manuscript.

Kind regards,

Michael Schubert

Academic Editor

PLOS ONE

Journal Requirements:

Reviewers' comments:

Reviewer's Responses to Questions

**Comments to the Author**

1. Is the manuscript technically sound, and do the data support the conclusions?

Reviewer #1: Partly

Reviewer #2: No

2. Has the statistical analysis been performed appropriately and rigorously? 

Reviewer #1: Yes

Reviewer #2: Yes

3. Have the authors made all data underlying the findings in their manuscript fully available?

Reviewer #1: Yes

Reviewer #2: Yes

4. Is the manuscript presented in an intelligible fashion and written in standard English?

Reviewer #1: Yes

Reviewer #2: Yes

5. Review Comments to the Author

Reviewer #1: This papers provides a very useful in depth actualisation of the NR2F repertoire, especially in vertebrates. However, at the eumetazoan scale, it it less extensive, and thus in the present state leads to less robust conclusions.

The weakest point is the data substantiating the author's hypothesis that NR2F6 was lost in most bilaterians but retained only in gnathostomes. Actually, this may be true but should be backed by more extensive sampling of NR2F sequences in cnidarians. The minimum would be to incorporate all the sequences previousely discussed as cnidarian NR2Fs, so four, not only three N. vectensis paralogs, and the sequences from H. vulgaris (mentionned in the text) and from A. millepora. Those were already incorporated in the Bridgham et al Plos Biology paper form 2010. See Supp Fig 2. More recently, an additional sequence from the hydrozoan Hydractinia echinata has been incorporated into a global NR analysis (Beinsteiner et al., Plos Genetics, 2021, Supp Fig 3), suggesting that this sequence belongs also to the NR2F group. I am pretty sure the authors could significantly improve the cnidarian dataset just by doing targeted blast queries on cnidarian sequences available in Genbank. With that they would probably get the data necessarily to back their hypothesis more solidely.

Another problem is that some sequences discussed in the text (like the nematode UNC-55) and the three hagfish sequences are missing. The hagfish ones should be present in the tree to demonstrate their orthology to the lamprey ones. Regarding the UNC-55, I can guess that the author were embarrassed by its unstable placing. According to Bertrand et al., MBE 2004, figure 1, this sequence did not branch clearly with the drosophila SVP/NR2F3. This is probably a long-branch artefact but could be easily resolved incorporating other nematode sequences like the one from Brugia malayi (EDP32461.1). Perhaps the authors could even found a Priapulid sequence that may help group together nematodes and arthropods.

Regarding nomenclature, it would be preferable to keep the name NR2F3 for the group the author call here NR2F1/2/5/6 should be named NR2F3, regarding the fact that the SVP Drosophila sequence was used to define that group in the official nomenclature paper of 1999 (doi:10.1016/s0092-8674(00)80726-6). At least for the ecdysozoan (or even protostome sequences) that are orthologous. Also, it should be mentionned somewhere in the paper that the xenopus sequence that the authors put in the NR2F5 group was initially named NR2F4. Actually, the coelacanth sequence could be also named NR2F4, being closer to the xenopus one than to the teleost one. And in that case, a name like NR2F4/5 would be appropriate for the entire gnathostome group. In any case, for all readers that are not familiar with the nomenclature, it would be useful to add trivial names, for sequences that have already been functionally characterized.

Regarding the rooting of the tree, the T. adhaerens sequence is not strickly speaking a NR2F. Its weak branching at the basis of the NR2F in the global trees of Bridgham and Beinsteiner indicates that it could be homologous to both NR2F and NR2E families in eumetazoans. Therefore it has been called NR2I in the Beinsteiner paper (see Fig 7). Based on this interpretation, I recommend rooting the tree using this sequence instead of the full cnidarian/placozoan group during the next analysis run.

Fig3: the comparison is unsufficiently exploited here. The authors should clearly highlight the colored position in the zinc fingers that are also present in Nematostella and Trichoplax. For example, the turquoise L residue from X. tropicalis is also present in T. adhaerens. The blue V is present in all thre N. vectensis and in the T. adharens sequences. The magenta G is present in two N. vectensis sequences. The green D is present in one N. vectensis sequence. An then, are those changes linked of functional significance? Did the author look at structural data related to the DBD?

Regarding the discussion of ancestral functions, it is hard to follow as currently written. I would recommend that the author implement a caracter mapping approach using the phytools package in R, mesquite or whatever program they would be comfortable with. This would once again give more solidity to their claims regarding the ancestrality of nervous system expression of NR2F genes.

Typos:

L64: which two contains two Zinc finger motifs -> which contains

L76: Kruppel-like 9 -> Krüppel-like factor 9

Reviewer #2: In this manuscript Coppola and Waxman address the evolution of NR2F transcription factors in this manuscript. This is an important group of nuclear receptors, which display a complex evolutionary history. The authors have made an effort to elucidate some outstanding questions regarding the evolution of these genes. I raise below a few questions that deserve attention prior to acceptance.

1. The authors refer NR2F5 absent from “aminotes” – confirm, as am I sure that there are orthologues in bird and reptile genomes. This needs clarification.

2. The authors consider NR2F originating “pre-metazoan” (line 8)– this cannot be. NRs emerged in the ancestor of Metazoa (check and cite https://doi.org/10.1371/journal.pbio.1000497). Clarify conclusions. NR2F emerged in the ancestor of Bilateria. Not including Porifera and others groups does not allow the sort of conclusions described in the manuscript.

3. NR2F6: unclear why the authors suggest this have been lost independently in so many lineages (figure 9; a more parsimonious scenario would be that NR2f6 is a highly divergent paralogue from 2R genome duplications? It would be relevant to investigate the paralogy relationships between these regions).

4. NR2F5 in cartilaginous fish: maybe consider sampling more species using a synteny based approach. Use Bola1 as an anchor gene? Sample more genomes available and readdress this conclusion. Please have a look and cite: DOI: 10.1016/j.ygcen.2020.113527

5. “Pre-metazoan models Amphimedon queenslandica (sponge) and Mnemiopsis leidyi (ctenophore)”: you mean pre Bilateria? Correct throughout the manuscript.

6. PLOS authors have the option to publish the peer review history of their article (what does this mean?). If published, this will include your full peer review and any attached files.

Reviewer #1: No

Reviewer #2: No

---

## [Author Response · Author response to Decision Letter 0]

28 Sep 2021

Response to Reviewers: PONE-D-21-18033

We thank both reviewers for their thoughtful and insightful critiques of our manuscript. Both reviewers recognized the importance of understanding the evolution of Nr2f transcription factors. While supportive of the work, they both provided helpful suggestions for additional analysis that would improve our study. In particular, these questions centered around the relationship of cnidarian Nr2fs and the Nr2f6 family within the phylogenetic analyses, and the presence of Nr2f5 orthologues within vertebrates. In the revised manuscript, we have tried to directly address all the reviewers’ comments. The new analyses performed in response to the reviewers’ comments have helped to resolve discrepancies within the original manuscript and dramatically improved our study. Responses to the reviewers’ specific critiques are indicated below in blue. Changes made within the revised manuscript in response to the reviewers’ comments are indicated in the manuscript version with track changes.

Reviewer #1: This papers provides a very useful in depth actualisation of the NR2F repertoire, especially in vertebrates. However, at the eumetazoan scale, it it less extensive, and thus in the present state leads to less robust conclusions.

1. The weakest point is the data substantiating the author's hypothesis that NR2F6 was lost in most bilaterians but retained only in gnathostomes. Actually, this may be true but should be backed by more extensive sampling of NR2F sequences in cnidarians. The minimum would be to incorporate all the sequences previousely discussed as cnidarian NR2Fs, so four, not only three N. vectensis paralogs, and the sequences from H. vulgaris (mentionned in the text) and from A. millepora. Those were already incorporated in the Bridgham et al Plos Biology paper form 2010. See Supp Fig 2. More recently, an additional sequence from the hydrozoan Hydractinia echinata has been incorporated into a global NR analysis (Beinsteiner et al., Plos Genetics, 2021, Supp Fig 3), suggesting that this sequence belongs also to the NR2F group. I am pretty sure the authors could significantly improve the cnidarian dataset just by doing targeted blast queries on cnidarian sequences available in Genbank. With that they would probably get the data necessarily to back their hypothesis more solidely.

As Reviewer #1 suggested, we have included additional cnidarian Nr2f sequences. We found that there are 4 cnidarian Nr2fs in N. vectensis and A. millepora, and 3 in H. vulgaris. We could not identify additional H. echinate Nr2fs based on available data. We found that the H. vulgaris, H. echinata, and an A. millepora Nr2f were significantly divergent, which caused long branch artifacts in the phylogenetic tress. Therefore, the analysis shown has excluded those. Nevertheless, the incorporation of the additional cnidarian sequences has improved our analysis of Nr2fs within cnidaria and the overall phylogenetic trees, and support that the additional Nr2fs in cnidaria originated from cnidaria-specific duplication events. The additional data is reported in the revised Fig 2, S1 Fig, Fig 5, on lines 120-127 of the revised manuscript, and included in the revised Abstract.

Regarding the Nr2f6 origin hypothesis reported in the original manuscript, we agree that it may have been confusing and needed greater support. However, it was based on our data at the time. In addition to the incorporation of additional cnidaria sequences, we included more vertebrate Nr2f5 sequences, and rooted the trees with the placozoan Nr2f, as was suggested. While we have not determined the individual effects of each these changes, the new analysis supports a Nr2f5/Nr2f6 branch, suggesting that Nr2f5 and Nr2f6 are paralogous, which is a more parsimonious evolutionary hypothesis and consistent with whole genome duplications in vertebrates. Furthermore, additional analysis of synteny revealed the presence of shared genes (Mef2 TFs) between Nr2f1/2 and Nr2f6 loci of vertebrates (see other reviewer’s comments) and that cnidarian Nr2f genes are flanked by Mef2, Arrdc, and Rbm8 homologs, which are found flanking the vertebrate Nr2f genes. Thus, the incorporation of new sequence data and rooting supports the orthology between cnidarian Nr2f1/2/5/6 and vertebrates’ Nr2fs, and that Nr2f6 is derived from genome duplication events in vertebrates (see revised Fig 2, S1 Fig, Fig 5, and Fig 8, lines 157-171 and 309-312, and included in the revised Abstract).

2. Another problem is that some sequences discussed in the text (like the nematode UNC-55) and the three hagfish sequences are missing. The hagfish ones should be present in the tree to demonstrate their orthology to the lamprey ones. Regarding the UNC-55, I can guess that the author were embarrassed by its unstable placing. According to Bertrand et al., MBE 2004, figure 1, this sequence did not branch clearly with the drosophila SVP/NR2F3. This is probably a long-branch artefact but could be easily resolved incorporating other nematode sequences like the one from Brugia malayi (EDP32461.1). Perhaps the authors could even found a Priapulid sequence that may help group together nematodes and arthropods.

As Reviewer suggested, we have incorporated the hagfish Nr2fs in the revised tree (see revised Fig 2). However, there is still limited genomic information to perform syntenic analysis with hagfish. We have incorporated synteny analysis for the one hagfish Nr2f where were able to find genomic information.

Yes, there were a few cases where we observed long branch artifacts created by significantly divergent sequences, with examples including some cnidaria (as mentioned above) and nematodes. We tried to be transparent about this in the original text and stated this in the original Methods. Sequences that were not included in the phylogenetic trees due to their high divergence and that caused long branch artifacts were presented in the original S5 File (revised S2 File). We did take Reviewer #1’s suggestion regarding the incorporation of additional nematode sequences (B. malayi, O. vulvusls), as well as used C. elegans, to try to alleviate these issues. However, the incorporation of these sequences always caused long branch artifacts, and we consequently excluded. We were able to include a Priapulid sequence. Thus, these new results are present in the revised phylogenetic trees (Fig 2 and S1 Fig).

3. Regarding nomenclature, it would be preferable to keep the name NR2F3 for the group the author call here NR2F1/2/5/6 should be named NR2F3, regarding the fact that the SVP Drosophila sequence was used to define that group in the official nomenclature paper of 1999 (doi:10.1016/s0092-8674(00)80726-6). At least for the ecdysozoan (or even protostome sequences) that are orthologous. Also, it should be mentionned somewhere in the paper that the xenopus sequence that the authors put in the NR2F5 group was initially named NR2F4. Actually, the coelacanth sequence could be also named NR2F4, being closer to the xenopus one than to the teleost one. And in that case, a name like NR2F4/5 would be appropriate for the entire gnathostome group. In any case, for all readers that are not familiar with the nomenclature, it would be useful to add trivial names, for sequences that have already been functionally characterized.

We agree with Reviewer #1 that within the manuscript and as a community we should incorporate proper nomenclature for Nr2fs and nuclear receptors in general. However, it is not clear to us that Nr2f3 and Nr2f4 that were suggested from the nomenclature committee >20 years ago are suitable names based in current analysis. These names were based on significantly fewer sequences from Nr2f members and we cannot even access their data based on the outdated links in that paper. Moreover, those suggested names have not really been adopted within the literature. We propose the names should reflect the proper homology of the genes, which is why we have incorporated Nr2f1/2/5/6 and suggest potentially Nr2f within the revised text. In the revised manuscript, we have tried to explain the rationale for our nomenclature proposal more thoroughly (lines 132-139 of revised manuscript). In addition, we have mentioned the previous names svp and Nr2f3 to alleviate any confusion. We have also mentioned that previous groups referred to Nr2f4 in Xenopus, as was suggested (lines 166-168 of revised manuscript). Again, we do not think there is rationale for this name based on the evolutionary analysis. It would seem to be unnecessarily confusing to incorporate this name for reptiles and coelacanth moving forward and think the names based on the relationships shown here should be corrected.

4. Regarding the rooting of the tree, the T. adhaerens sequence is not strickly speaking a NR2F. Its weak branching at the basis of the NR2F in the global trees of Bridgham and Beinsteiner indicates that it could be homologous to both NR2F and NR2E families in eumetazoans. Therefore it has been called NR2I in the Beinsteiner paper (see Fig 7). Based on this interpretation, I recommend rooting the tree using this sequence instead of the full cnidarian/placozoan group during the next analysis run.

Thank you for the suggestions. As mentioned above, we did root the revised phylogenetic trees with placozoan Nr2f. However, we respectfully disagree with presently calling the placozoan “Nr2f” sequence used “Nr2I” based on the available data and analysis. In Beinsteiner et al, there is really no discussion of the rationale for the use of this name in the manuscript. However, in the phylogenetic trees presented in both Bridgham et al and Beinsteiner et al, the placozoan Nr2f gene segregates with other Nr2f genes and is not found at the base of Nr2f and Nr2e genes, which would provide greater evidence for it being “Nr2I.” Moreover, sequence alignments shows that the placozoan Nr2f protein is more highly conserved, sharing ~10% greater identity and ~30% similarity at the sequence level, with Nr2fs than Nr2fe in vertebrates, including key residues found in all Nr2f proteins. Additionally, as shown in this manuscript, the intron/exon structure of the placozoan Nr2f is conserved with invertebrate and vertebrate Nr2fs (Fig 4 and lines 211-212). This intron/exon structure is not conserved with Nr2e genes. Therefore, we think the present data support that the placozoan Nr2f sequence used should be named called “Nr2f” and not “Nr2I.”

5. Fig3: the comparison is unsufficiently exploited here. The authors should clearly highlight the colored position in the zinc fingers that are also present in Nematostella and Trichoplax. For example, the turquoise L residue from X. tropicalis is also present in T. adhaerens. The blue V is present in all thre N. vectensis and in the T. adharens sequences. The magenta G is present in two N. vectensis sequences. The green D is present in one N. vectensis sequence. An then, are those changes linked of functional significance? Did the author look at structural data related to the DBD?

We have revised Fig 3 to include more sequences and highlight the appropriate residues found throughout all the protein sequences shown (see revised Fig 3 and Fig 3 legend). Although there is quite a lot of structure information about LBDs, there is significantly less about Nr2f structures in DBD. Thus, we are not aware of structural information about the DBDs that we could use to make meaningful predictions about function. We have stated this in the revised text (lines 186-188).

6. Regarding the discussion of ancestral functions, it is hard to follow as currently written. I would recommend that the author implement a caracter mapping approach using the phytools package in R, mesquite or whatever program they would be comfortable with. This would once again give more solidity to their claims regarding the ancestrality of nervous system expression of NR2F genes.

Thank you for the suggestion. We apologize our discussion was confusing. While character mapping is certainly one approach that can be used to decipher hypothesis about ancestral function, it was not clear this analysis really added much given the limited data in a variety of models. Therefore, in the revised manuscript we tried to address this issue by clarifying and simplifying the Discussion regarding the ancestral functions of Nr2f genes within neural tissues based on current data (page lines 402-430).

7. Typos:

L64: which two contains two Zinc finger motifs -> which contains

L76: Kruppel-like 9 -> Krüppel-like factor 9

Thank you. We corrected these typos.

Reviewer #2: In this manuscript Coppola and Waxman address the evolution of NR2F transcription factors in this manuscript. This is an important group of nuclear receptors, which display a complex evolutionary history. The authors have made an effort to elucidate some outstanding questions regarding the evolution of these genes. I raise below a few questions that deserve attention prior to acceptance.

1. The authors refer NR2F5 absent from “aminotes” – confirm, as am I sure that there are orthologues in bird and reptile genomes. This needs clarification.

In the revised manuscript, we have tried to clarify the losses of Nr2f5 found in amniotes. Examining the genomes of additional vertebrate species, we have found Nr2f5 in some reptiles (alligators, lizards, and turtles). However, it also appears to be lost in some turtles (i.e. the soft-shell turtle) and we could not find it in any bird and mammalian genomes examined. Thus, the most parsimonious hypothesis is that Nr2f5 has had independent losses in some amniote lineages. In the revised manuscript, we have incorporated the additional analysis (revised Fig 2) and tried to clarify this point (lines 159-166 and lines 384-386). We have also provided a list of reptile, bird, and mammalian genomes in which we were not able to find Nr2f5 genes in the Supporting Information (revised S3 File).

2. The authors consider NR2F originating “pre-metazoan” (line 8)– this cannot be. NRs emerged in the ancestor of Metazoa (check and cite https://doi.org/10.1371/journal.pbio.1000497). Clarify conclusions. NR2F emerged in the ancestor of Bilateria. Not including Porifera and others groups does not allow the sort of conclusions described in the manuscript.

Thank you for pointing this out. We agree and apologize that our statements were confusing. Throughout the revised manuscript we have eliminated the term “pre-metazoan” and have tried to clarify the statements about the origins of the Nr2fs in metazoa and their evolution in eumetazoa.

3. NR2F6: unclear why the authors suggest this have been lost independently in so many lineages (figure 9; a more parsimonious scenario would be that NR2f6 is a highly divergent paralogue from 2R genome duplications? It would be relevant to investigate the paralogy relationships between these regions).

We agree that our previous hypothesis was not parsimonious based on the 2R genome duplications. We have now included additional sequences in our analysis, including more cnidaria Nr2fs and vertebrate Nr2f5 sequences, and rooted the phylogenetic trees with the placozoan Nr2f (see response to Reviewer #1, comment 1). This seems to have resolved this issue and our phylogenetic trees now supports the parsimonious scenario of a 2R-origin for Nr2f6 and that it is paralogous to Nr2f5. Furthermore, additional analysis of synteny shows that several Nr2f6 loci are flanked by Mef2b, while paralogues Mef2c and –a flank Nr2f1 and Nr2f2, respectively. These new analyses are incorporated in the text (lines 157-171 and 309-320), the revised phylogenetic trees (Fig 2 and S1 Fig), the revised synteny schematic (Fig 8), and mentioned in the revised Abstract. 

4. NR2F5 in cartilaginous fish: maybe consider sampling more species using a synteny based approach. Use Bola1 as an anchor gene? Sample more genomes available and readdress this conclusion. Please have a look and cite: DOI: 10.1016/j.ygcen.2020.113527

Thank you. We have sampled additional cartilaginous species. We found that Nr2f5 is present in the Great white shark (C. carcharias) and incorporated the Nr2f5 sequence for the Whale shark reported in Fonseca et al. However, we were not able to identify Nr2f5 orthologues in ghost shark, catshark, and skates (S3 Fig), suggesting it had been independently lost in some cartilaginous fish. Importantly, the Great white shark genome allowed us to also perform syntenic analysis of the loci using Bola1 as an anchor as was suggested. These data are incorporated in the revised manuscript in revised Fig 2, S1 Fig, and Fig 7, and lines 159-166.

5. “Pre-metazoan models Amphimedon queenslandica (sponge) and Mnemiopsis leidyi (ctenophore)”: you mean pre Bilateria? Correct throughout the manuscript.

Thank you. Yes, we referred to those animals incorrectly. We have corrected this mistake throughout the manuscript.

---

## [Decision Letter · Decision Letter 1]

8 Nov 2021

Origin and evolutionary landscape of Nr2f transcription factors across Metazoa

PONE-D-21-18033R1

Dear Dr. Waxman,

We’re pleased to inform you that your manuscript has been judged scientifically suitable for publication and will be formally accepted for publication once it meets all outstanding technical requirements.

Kind regards,

Michael Schubert

Academic Editor

PLOS ONE

Additional Editor Comments (optional):

Reviewers' comments:

Reviewer's Responses to Questions

**Comments to the Author**

1. If the authors have adequately addressed your comments raised in a previous round of review and you feel that this manuscript is now acceptable for publication, you may indicate that here to bypass the “Comments to the Author” section, enter your conflict of interest statement in the “Confidential to Editor” section, and submit your "Accept" recommendation.

Reviewer #1: All comments have been addressed

Reviewer #2: All comments have been addressed

2. Is the manuscript technically sound, and do the data support the conclusions?

Reviewer #1: Yes

Reviewer #2: Yes

3. Has the statistical analysis been performed appropriately and rigorously? 

Reviewer #1: Yes

Reviewer #2: Yes

4. Have the authors made all data underlying the findings in their manuscript fully available?

Reviewer #1: Yes

Reviewer #2: Yes

5. Is the manuscript presented in an intelligible fashion and written in standard English?

Reviewer #1: Yes

Reviewer #2: Yes

6. Review Comments to the Author

Reviewer #1: The authors addressed all requests in a satisfactory manner. The manuscript is therefore suitable for publication now.

Reviewer #2: The authors have addressed my comments in the first round of revision. Thus, I support the publication of this manuscript.

7. PLOS authors have the option to publish the peer review history of their article (what does this mean?). If published, this will include your full peer review and any attached files.

Reviewer #1: No

Reviewer #2: No

---

## [Editor Report · Acceptance letter]

12 Nov 2021

PONE-D-21-18033R1 

Origin and evolutionary landscape of *Nr2f* transcription factors across Metazoa 

Dear Dr. Waxman:

I'm pleased to inform you that your manuscript has been deemed suitable for publication in PLOS ONE. Congratulations! Your manuscript is now with our production department. 

Kind regards, 

on behalf of

Dr. Michael Schubert 

Academic Editor

PLOS ONE